# Synthetic reconstruction of the *hunchback* promoter specifies the role of Bicoid, Zelda and Hunchback in the dynamics of its transcription

Gonçalo Fernandes[1†], Huy Tran[1,2†], Maxime Andrieu[1], Youssoupha Diaw[1], Carmina Perez Romero[3], Cécile Fradin[3,4], Mathieu Coppey[5], Aleksandra M Walczak[2]*, Nathalie Dostatni[1]*

[1]Institut Curie, Université PSL, Sorbonne Université, CNRS, Nuclear Dynamics, Paris, France; [2]Laboratoire de Physique de l'École Normale Supérieure, CNRS, Université PSL, Sorbonne Université and Université de Paris, Paris, France; [3]Department of Biochemistry and Biomedical Sciences, McMaster University, Hamilton, Canada; [4]Department of Physics and Astronomy, McMaster University, Hamilton, Canada; [5]Institut Curie, Université PSL, Sorbonne Université, CNRS UMR168, Laboratoire Physico Chimie Curie, Paris, France

*For correspondence:
aleksandra.walczak@phys.ens.fr (AMW);
nathalie.dostatni@curie.fr (ND)

†These authors contributed equally to this work

**Abstract** For over 40 years, the Bicoid-*hunchback* (Bcd-*hb*) system in the fruit fly embryo has been used as a model to study how positional information in morphogen concentration gradients is robustly translated into step-like responses. A body of quantitative comparisons between theory and experiment have since questioned the initial paradigm that the sharp *hb* transcription pattern emerges solely from diffusive biochemical interactions between the Bicoid transcription factor and the gene promoter region. Several alternative mechanisms have been proposed, such as additional sources of positional information, positive feedback from Hb proteins or out-of-equilibrium transcription activation. By using the MS2-MCP RNA-tagging system and analysing in real time, the transcription dynamics of synthetic reporters for Bicoid and/or its two partners Zelda and Hunchback, we show that all the early *hb* expression pattern features and temporal dynamics are compatible with an equilibrium model with a short decay length Bicoid activity gradient as a sole source of positional information. Meanwhile, Bicoid's partners speed-up the process by different means: Zelda lowers the Bicoid concentration threshold required for transcriptional activation while Hunchback reduces burstiness and increases the polymerase firing rate.

## Editor's evaluation

In this paper, the authors use synthetic transcriptional enhancers to probe the roles of three transcription factors, Bicoid, Hunchback and Zelda, in specifying the production of a sharp, accurately placed gene expression boundary in early fruit fly embryos. They find that Bicoid, which is expressed in an anterior-posterior gradient, is sufficient on its own to generate a boundary in the same location as a wild-type fly, but combinatorial regulation by Hunchback and Zelda is needed to ensure the boundary forms quickly enough. They further combine their experimental observations with modeling to conclude that Bicoid exists in active and inactive forms, and that an equilibrium model captures the relevant behaviors, implying energy expenditure during the binding of transcription factors to DNA or RNA polymerase is theoretically unnecessary.

## Introduction

Morphogen gradients are used by various organisms to establish polarity along embryonic axes or within organs. In these systems, positional information stems from the morphogen concentration detected by each cell in the target tissue and mediates the determination of cell identity through the expression of specific sets of target genes. While these processes ensure the reproducibility of developmental patterns and the emergence of properly proportioned individuals, the question of whether the morphogen itself directly contributes to this robustness or whether it requires the involvement of downstream cross-regulatory networks or cell-communication remains largely debated. This question becomes even more pressing with the recent discovery that when studied at the single-cell level, transcription is frequently observed to be an extremely noisy process, hardly suggestive of such precise control.

To understand how reproducible transcription patterns can robustly emerge from subtle differences of morphogen concentration, we study the Bicoid (Bcd) morphogen system which initiates pattern formation along the antero-posterior (AP) axis in the young fruit fly embryo (*Driever and Nüsslein-Volhard, 1988*). The Bcd gradient was shown to be steadily established at the onset of transcription, one hour after egg laying, in the form of an exponential AP gradient with a $\lambda$ decay length measured in the range of 16–25% egg-length (EL) (*Abu-Arish et al., 2010*; *Durrieu et al., 2018*; *Houchmandzadeh et al., 2002*; *Liu et al., 2013*). Fluorescent correlation spectroscopy measurements (*Abu-Arish et al., 2010*) and single molecule tracking of GFP-tagged Bcd proteins (*Mir et al., 2018*) revealed that a fraction of the Bcd proteins has a fast diffusion coefficient sufficient to explain the establishment of the gradient in such a short time by the synthesis-diffusion-degradation model (*Abu-Arish et al., 2010*; *Fradin, 2017*). This was further supported with the use of a tandem fluorescent timer as a protein age sensor (*Durrieu et al., 2018*). Of note, the establishment of the Bcd gradient is not only rapid but also extremely precise in space with only 10% variability among embryos (*Gregor et al., 2007b*) and the gradient is linearly correlated to the amount of *bcd* mRNA maternally provided and the number of functional *bcd* alleles in the females (*Liu et al., 2013*; *Petkova et al., 2014*).

The Bcd protein binds DNA through its homeodomain (*Hanes and Brent, 1989*; *Treisman et al., 1989*) and activates the expression of a large number of target genes carrying Bcd binding sites (BS). Among the Bcd target genes, *hunchback* (*hb*) is expressed in a large domain spanning the whole anterior half of the embryo (*Driever et al., 1989*). *hb* expression begins when the first hints of transcription are detected in the embryo, i.e. at nuclear cycle 8 (*Porcher et al., 2010*). About one hour later (i.e. at nuclear cycle 14), the expression domain of *hb* is delimited by a posterior boundary, which is both precisely positioned along the AP axis and very steep suggesting that very subtle differences in Bcd concentration in two nearby nuclei at the boundary are already precisely measured to give rise to very different transcriptional responses (*Crauk and Dostatni, 2005*; *Gregor et al., 2007a*; *Houchmandzadeh et al., 2002*). Detailed analysis of *hb* expression by RNA FISH also indicated that transcription at the *hb* locus is extremely dynamic in time: it is detected during the successive S-phases but not during the intervening mitoses, which punctuate this period of development.

To gain insights into the dynamics of *hb* early expression with a higher temporal resolution, the MS2-MCP approach for fluorescent tagging of RNA (*Ferraro et al., 2016*) was adapted to living fruit fly embryos (*Lucas et al., 2013*; *Garcia et al., 2013*). This provided an hb-P2 MS2-reporter expressed anteriorly in a domain with a boundary of the same steepness and positioning precision as the endogenous *hb* (*Lucas et al., 2018*). Of note, despite, this highly reproducible measurement of positional information (position and steepness of the boundary) on the scale of the embryo, at the single locus level, the variability in the total mRNA production (δmRNA/mRNA) over an entire nuclear cycle for loci at the boundary was of 150 %, i.e. one locus can produce 2.5 X more mRNA than another locus (*Desponds et al., 2016*). This high variability (noise) was consistent with smFISH data measuring the variability of *hb* mRNA amounts in nuclei (*Little et al., 2013*). It reflects a stochastic transcription process in neighboring nuclei which nevertheless all make the precise decision to turn ON *hb* during the cycle.

The transcription dynamics of the hb-P2 MS2-reporter indicated that its steep boundary is established at each nuclear cycle 11–13 within 180 s and therefore suggested that accurate measurements of Bcd concentration were made much more rapidly than anticipated (*Lucas et al., 2018*). Consistently, inactivating Bcd by optogenetics in the embryo indicated that the *hb* transcription exhibited a very fast sensitivity to Bcd activity (*Huang et al., 2017*). Modeling was used to recapitulate the

observed dynamics assuming cooperative binding of Bcd proteins to the six known BS sites of the hb-P2 promoter and rate limiting concentrations of Bcd at the boundary (*Tran et al., 2018a*). The model was able to recapitulate the fast temporal dynamics of the boundary establishment but could not reproduce its observed steepness which, given the 20% EL decay length of the Bcd protein gradient measured with immuno-staining (*Houchmandzadeh et al., 2002*), corresponds to a Hill coefficient of ~7, difficult to achieve without invoking the need for additional energy expenditure (*Estrada et al., 2016*). As expected, the performance of the model was higher when increasing the number of Bcd BS above six with a minimum of 9 Bcd BS required to fit the experimental data with a boundary of the appropriate steepness. This indicated that either the hb-P2 promoter contained more than 6 Bcd BS or that additional mechanisms were required to account for the steepness of the boundary.

While quantitative models based on equilibrium binding of transcription factors to DNA shed lights on segmentation in *Drosophila* (*Segal et al., 2008*) or on the Bcd system (*Estrada et al., 2016*; *Tran et al., 2018a*), their impact remained limited by the lack of a quantitative experimental systems for validation. Here, we combined the MS2 quantitative probing system with a synthetic approach to decipher the functioning of Bcd in the transcription process at the mechanistic level. We built Bcd-only reporters with specific numbers of Bcd BS as well as reporters with 6 Bcd BS in combination with BS for the two known maternal Bcd co-factors binding to the hb-P2 promoter, namely the Hb protein itself (*Porcher et al., 2010*; *Simpson-Brose et al., 1994*) and the Zelda (Zld) pioneer transcription factor (*Hannon et al., 2017*; *Xu et al., 2014*). We show that 6 Bcd BS are not sufficient to recapitulate the hb-P2 expression dynamics while a reporter with only 9 Bcd BS recapitulates most of its spatial features, except a slightly lower steepness of its expression boundary and a longer period to reach steady state. To account for the bursty behavior of Bcd-only reporters in excess of Bcd, we fitted our data to a model involving a first step of Bcd binding/unbinding to the BS array and a second step where the bound Bcd molecules activate transcription. Synthetic reporters combining Bcd BS with either Hb or Zld BS indicated that both Hb and Zld sites reduce the time to reach steady state and increase expression by different means: Zld sites contribute to the first step of the model by drastically lowering the Bcd concentration thresholds required for activation while Hb sites act in the second step by reducing Bcd-induced burstiness and increasing the polymerase firing rates. Importantly, in embryos maternally expressing one (1 X) *vs* two (2 X) *bcd* functional copies, the boundary shift of the Bcd-only synthetic reporter with 9 Bcd BS was small enough to set the Bcd system within the limits of an equilibrium model. Lastly, the shift observed for the hb-P2 reporter in 1 X *vs* 2 X *bcd* backgrounds was the same as for the synthetic reporters further supporting that the Bcd gradient is the main source of positional information for the early expression of *hb*.

## Results

### Nine Bicoid binding sites alone recapitulate most features of the hb-P2 pattern

We first investigated the transcription dynamics of Bcd-only MS2 reporters carrying exclusively 6, 9, or 12 strong Bcd binding sites (BS) (*Hanes and Brent, 1989*; *Treisman et al., 1989*) upstream of an hsp70 minimal promoter (*Figure 1A* and *Supplementary file 1*), all inserted at the same genomic location (see Materials and methods and *Figure 1—figure supplement 1*). Videos were recorded (see *Videos 1–3*) and analyzed from nuclear cycle 11 (nc11) to 13 (nc13) but we focused on nc13 data, which are statistically stronger given the higher number of nuclei analyzed. Unless otherwise specified, most conclusions were also valid for nc11 and nc12. Given that the insertion of a BAC spanning the whole endogenous *hb* locus with all its Bcd-dependent enhancers did not affect the regulation of the wild-type gene (*Lucas et al., 2018*), it is unlikely that there will be competition for Bcd binding between the endogenous *hb* and these synthetic reporters.

The expression of the B6 (6 Bcd BS), B9 (9 Bcd BS), and B12 (12 Bcd BS) reporters harbored similar features as expression of the hb-P2 reporter (*Lucas et al., 2018*), which carries the ~300 bp of the hb-P2 promoter and the *hb* intron (*Figure 1B*, *Supplementary file 1*, *Video 4*): during the cycle, transcription was first initiated in the anterior with the expression boundary moving rapidly toward the posterior to reach a stable position into nc13 (*Figure 1C*). For all synthetic reporters, the earliest time when transcription was detected following mitosis (averaged over nuclei at the same position, see also Materials and methods), $T_0$, showed a dependence on position along the AP axis (*Figure 1D*),

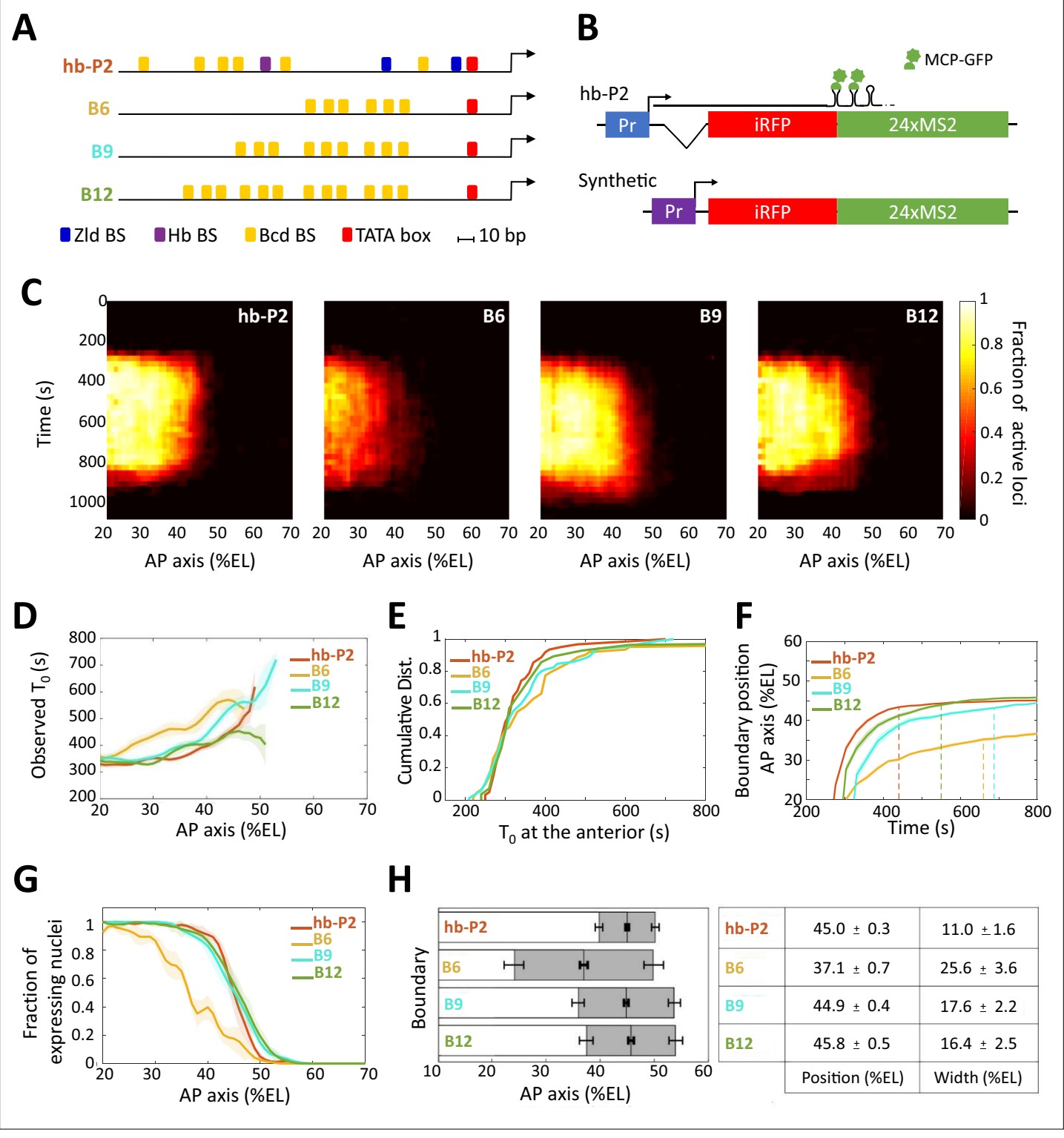

**Figure 1.** Transcription dynamics of the hb-P2, B6, B9 and B12 reporters. (**A**) Arrangement of the binding sites for Bcd (yellow), Hb (purple), and Zld (blue) upstream of the TATA box (red) and the TSS (broken arrow) of each reporter. (**B**) The MS2 reporters express the iRFP coding sequence followed by the sequence of the 24 MS2 stem loops. In the hb-P2 reporter, the hb-P2 promoter, 5'UTR sequence of the endogenous *hb* and its intron are placed just upstream of the iRFP sequence. In the synthetic reporters, the minimal promoter of the hsp70 gene was used. Of note, replacing the minimal promoter of hsp70 in B6 by the *hb* minimal promoter leads to a reporter with lower activity (*Figure 1—figure supplement 1*), (**F–G**). (**C**) Kymographs of mean fraction of active loci (colormap on the right) as a function of time (Y axis in s) and nuclei position along the AP axis (X axis in %EL) at nc13. (**D**) Along the

*Figure 1 continued on next page*

Figure 1 continued

AP axis (%EL), mean time of first spot appearance $T_0$ (s) with shaded standard error of the mean and calculated only for loci with observed expression. (E) Cumulative distribution function of $T_0$ (s) in the anterior (20% ± 2.5 %EL). (F) Boundary position (%EL) of fraction of nuclei with MS2 signal along AP axis, with shaded 95% confidence interval, as a function of time. The dash vertical lines represent the time to reach the final decision boundary position ( ± 2 %EL). (G) Fraction of nuclei with any MS2 signal, averaged over n embryos, with shaded standard error of the mean, along the AP axis (%EL), at nc13. (H) Boundary position and width were extracted by fitting the patterns fraction of expressing nuclei, (G) with a sigmoid function. Bar plots with 95% confidence interval for boundary position and width as the gray region placed symmetrically around the boundary position. Average values and confidence intervals are indicated in the adjacent table. (D–H) reporter data are distinguished by color: hb-P2 (orange, n = 5 embryos), B6 (yellow, n = 5 embryos), B9 (cyan, n = 6 embryos), and B12 (green, n = 4 embryos).

The online version of this article includes the following figure supplement(s) for figure 1:

**Figure supplement 1.** Comparison of transcription dynamics with hb-P2 reporters.

as observed for hb-P2 (*Lucas et al., 2018*). Thus, Bcd concentration is a rate-limiting factor for the expression of all reporters. As indicated by the distributions of onset time $T_0$ in the anterior (~20 %EL), the first transcription initiation time at high Bcd concentration were not statistically different (*P*-values > 0.5) for all synthetic reporters (B6, B9, or B12) and hb-P2 (*Figure 1E*). This contrasts to the middle of the axis where the absolute number of Bcd molecules has been evaluated to be around 700 (*Gregor et al., 2007a*) and where the Bcd protein is thus likely to be limiting: transcription dynamics of the various reporters was quite diverse (*Figure 1F*) and the time it took for the hb-P2 reporter to reach the final decision to position its boundary (converging time, *Supplementary file 2*) was only 225 ± 25 s while it took about twice as much time for B6 (425 ± 25 s) or B9 (475 ± 25 s) and slightly less for B12 (325 ± 25 s).

For all reporters, the fraction of nuclei with MS2 signal during the cycle exhibited a sigmoid-like pattern along the AP axis reaching 100% in the anterior and 0% in the posterior (*Figure 1G*). We fitted these patterns with sigmoid functions of position along the AP axis and extracted (see Materials and methods) quantitative values for the position and width of the expression boundary (*Figure 1H*). Increasing the number of Bcd BS from 6 to 9, shifted the expression boundary toward the posterior and decreased the width of the boundary (*Figure 1H*), whereas increasing the number of Bcd sites from 9 to 12 did not significantly change the boundary position nor the boundary width. Of note, B9, B12, and hb-P2 expression boundaries were at almost identical positions while the width of the hb-P2 boundary was smaller than the width of the B9 or the B12 boundaries (*Figure 1H*).

Thus, even though 6 Bcd BS have been described in the hb-P2 promoter, having only 6 Bcd BS alone in a synthetic reporter is not sufficient to recapitulate the *hb* pattern. Increasing this number up to nine is sufficient to recapitulate almost all spatial features of the hb-P2 pattern except for the steepness of the expression boundary. Of note, the Bcd-only reporters take much longer than the hb-P2 reporter to reach the final decision for boundary positioning suggesting that binding of additional transcription factors in the hb-P2 promoter likely contribute to speeding-up the process.

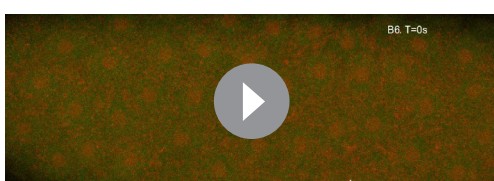

**Video 1.** Live imaging of transcription dynamics of B6 reporter. The videos have two channels: MCP-GFP channel (green) for monitoring the dynamics of nascent mRNA production and His-RFP (red) for nuclei detection. The capture frame is from 15% to 65% of embryo length. The anterior pole is on the left side of the frame. Position along the AP axis is indicated by white vertical bars positioned every 10% EL with the tallest one corresponding to 50% EL.

https://elifesciences.org/articles/74509/figures#video1

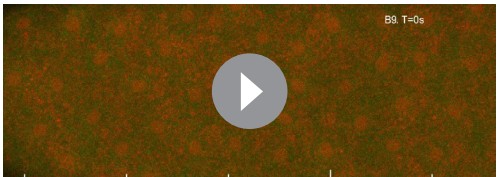

**Video 2.** Live imaging of transcription dynamics of B9 reporter. The videos have two channels: MCP-GFP channel (green) for monitoring the dynamics of nascent mRNA production and His-RFP (red) for nuclei detection. The capture frame is from 18% to 65% of embryo length. The anterior pole is on the left side of the frame. Position along the AP axis is indicated by white vertical bars positioned every 10% EL with the tallest one corresponding to 50% EL.

https://elifesciences.org/articles/74509/figures#video2

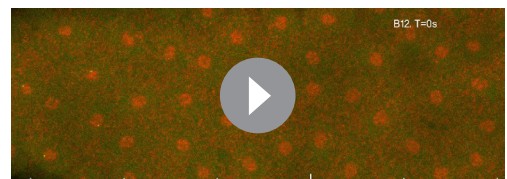

**Video 3.** Live imaging of transcription dynamics of B12 reporter. The videos have two channels: MCP-GFP channel (green) for monitoring the dynamics of nascent mRNA production and His-RFP (red) for nuclei detection. The capture frame is from 20% to 70% of embryo length. The anterior pole is on the left side of the frame. Position along the AP axis is indicated by white vertical bars positioned every 10% EL with the tallest one corresponding to 50% EL.

https://elifesciences.org/articles/74509/figures#video3

## Bicoid-dependent transcription is bursty at steady state even in excess of Bicoid

To study the kinetics of transcription induced by Bcd, we compared the dynamics of transcription of the hb-P2 and the Bcd-only reporters at steady state (in the time window of 600–800 s). From the time trace of MS2 activity in each nucleus, the fluctuation of the transcription process (burstiness, *Figure 2—figure supplement 1*) at a given position along the AP axis was featured by $P_{Spot}$, the average fraction of the cycle length during which fluorescent spots were observed (*Figure 2A*). In the anterior (~20 %EL), $P_{Spot}$ increased when increasing the number of Bcd BS in synthetic reporters from 6 to 9, with $P_{Spot}(B6) = 0.47 \pm 0.02$ and $P_{Spot}(B9) = 0.80 \pm 0.07$. $P_{Spot}(hb\text{-}P2) =$

0.84 ± 0.008 was as high as for B9 or B12 ($P_{Spot}(B12) = 0.76 \pm 0.07$). These values were all smaller than the fraction of expressing nuclei ( ~ 1, *Figure 1G*). This indicated bursty transcription activity in individual nuclei for all reporters, as confirmed by their individual MS2 traces in this region. Interestingly, $P_{Spot}$ for all Bcd-only reporters reach a plateau in the anterior where the Bcd concentration is in excess (*Figure 2A* and *Figure 2—figure supplement 1*). As in this region the Bcd BS on those reporters are likely to be always occupied by Bcd molecules, the burstiness observed is not caused by the binding/unbinding of Bcd to the BS array but by downstream processes. Meanwhile, the mean intensity of the MS2 signals ($\mu_I$) in the anterior region did not vary significantly (all p-value of KS test >0.07) between reporters (*Figure 2B*), suggesting that the number of bound Bcd molecules does not regulate the RNAP firing rate within transcription bursts.

## A model to recapitulate expression dynamics from Bicoid-only synthetic reporters

To explain the observed dynamics of the expression patterns (*Figure 1C*) and bursty transcription in regions with excess Bcd (*Figure 2A* and *Figure 2—figure supplement 1*), we built a model for transcription regulation of the Bcd-only synthetic reporters (*Figure 2*, C-D). In this model, regulation occurs in two steps: first, nuclear Bcd molecules can bind to and unbind from the Bcd BS on the promoter (*Figure 2C*) and second, bound Bcd molecules can activate transcription (*Figure 2D*). We assumed a static Bcd gradient, i.e. the Bcd concentration at a given position is constant over time. This was motivated by previous works on the dynamics of the intranuclear Bcd gradient using fluorescent-tagged Bcd at least during nc13 of our interest (*Abu-Arish et al., 2010*; *Gregor et al., 2007b*).

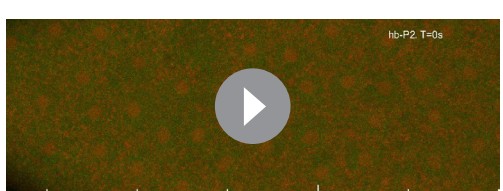

**Video 4.** Live imaging of transcription dynamics of hb-P2 reporter. The videos have two channels: MCP-GFP channel (green) for monitoring the dynamics of nascent mRNA production and His-RFP (red) for nuclei detection. The capture frame is from 15% to 70% of embryo length. The anterior pole is on the left side of the frame. Position along the AP axis is indicated by white vertical bars positioned every 10% EL with the tallest one corresponding to 50% EL.

https://elifesciences.org/articles/74509/figures#video4

In step 1 (*Figure 2C*), the binding and unbinding of Bcd to an array of $N$ identical Bcd BS were modeled explicitly, as in *Estrada et al., 2016*; *Tran et al., 2018a*. In our model, the binding state was denoted by $S_i$, with $i$ the number of bound Bcd molecules ($i \leq N$). The binding rate constants $k_i$ depend on the number of free BS ($N - i + 1$) and the Bcd search rate for a single BS $k_b$. The unbinding rate constants $k_{-i}$ were varied to account for various degrees of Bcd-DNA complex stability and binding cooperativity. In step 2 (*Figure 2D*), we expanded this model to account for the burstiness in transcription uncoupled with Bcd binding/unbinding (*Figure 2A*). The promoter dynamics was modeled as a two-state model, ON and OFF, to account for the observed bursts of transcription with a moderate time scale

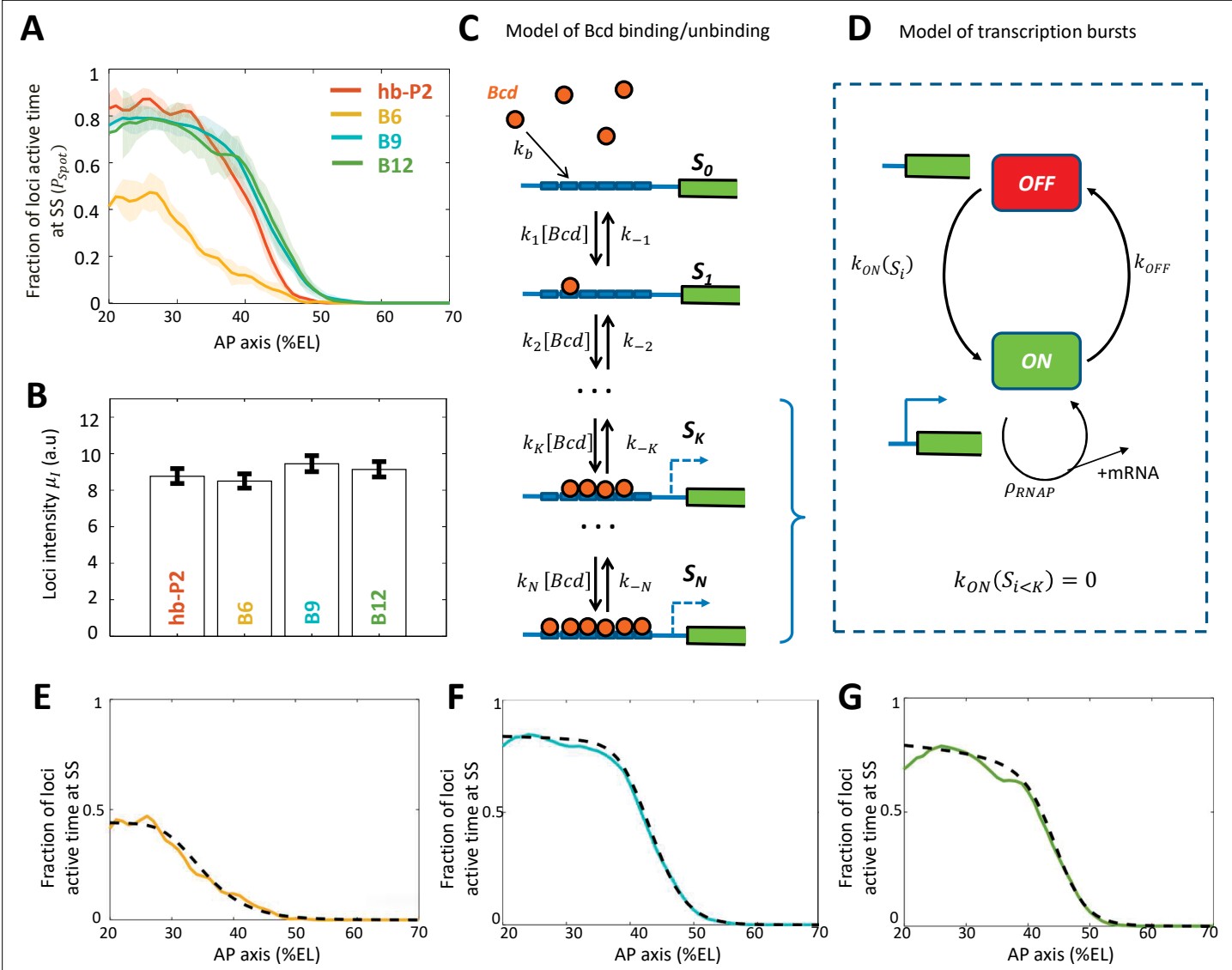

**Figure 2.** Modeling transcription dynamics at steady state. (**A**) Fraction of loci active time ($P_{Spot}$) at steady state (time window of 600–800 s into nc13), averaged over n embryos, as a function of nuclei position along AP axis (%EL). (**B**) Mean fluorescent intensity ($\mu_I$) with standard error of active MS2 loci detected in the anterior region ( ~ 20% ± 2.5% EL) at steady state. In (**A-B**) reporter data are distinguished by color: hb-P2 (orange, n = 5 embryos), B6 (yellow, n = 5 embryos), B9 (cyan, n = 6 embryos), and B12 (green, n = 4 embryos). (**C**) Model of Bicoid binding and unbinding to an array of $N$ identical binding sites: nuclear Bcd molecules can bind independently to individual binding sites at rate $k_b$. The binding array state is denoted by $S_i$ where $i$ is the number of bound sites. The forward rate constants $k_i$ are the binding rates of Bcd to the free remaining sites of $S_{i-1} : k_i = (N - i + 1)\,k_b$. The backward rate constants $k_{-i}$ are the unbinding rates of bound Bcd from $S_i$. (**D**) Transcription dynamics is modeled as a bursty two-state ON/OFF model with the switching rate constants $k_{ON}(S_i)$ and $k_{OFF}$. The switching rate $k_{ON}(S_i)$ depends on $i$ the number of bound Bcd molecules. Transcription is not activated with fewer than $K$ bound Bcd ($k_{ON}(S_{i<K}) = 0$). Only during the ON state can RNAPs arrive and initiate transcription at rate constant $\rho_{RNAP}$. (**E–G**) Fraction of active loci at steady state obtained experimentally for B6 (E, solid yellow), B9 (F, solid cyan), B12 (G, solid green) compared to the fraction of active loci at steady state from the best fitting models (dashed black) for corresponding BS numbers $N$=6 for B6 (**E**), $N$=9 for B9 (**F**) or N =12 for B12 (**G**). In these models, the free parameters are the unbinding rate constant ($k_{-i}$), the promoter switching rates with $K$ bound Bcd molecules ($k_{ON}(S_K)$ and $k_{OFF}$). $K$ is set to 3. The switching ON rates at higher bound states are set $k_{ON}(S_{i>K}) = k_{ON}(S_K)\left(\frac{i}{k}\right)$, given the synergistic activation of transcription by bound Bcd (see **Supplementary file 5**). The binding rate constant $k_b$ is determined by assuming that Bcd binding is diffusion limited (Appendix 2).

The online version of this article includes the following figure supplement(s) for figure 2:

**Figure supplement 1.** Example of timely transcription dynamics (bursting) monitored with MS2-MCP system.

**Figure supplement 2.** Fitting models to experimental data for the B9, B12, Z2B6, and H6B6 with the least degree of freedom when compared to B6.

**Figure supplement 3.** Schemes of transcription activation by bound Bcd molecules.

between 10 s and 100 s (*Bothma et al., 2014*; *Desponds et al., 2016*; *Lammers et al., 2020*). The turning ON rate $k_{ON}(S_i)$ was modulated by $i$ the number of bound Bcd molecules. When the BS arrays had less than $K$ Bcd molecules ($K \geq 0$), transcription could not be activated ($k_{ON}(S_{i<K}) = 0$). To account for the uncoupling between the burstiness of transcription and the Bcd binding and unbinding, the turning OFF rate $k_{OFF}$ did not depend on the Bcd BS state. When the promoter is ON, RNAP could initiate transcription and be fired at rate $\rho_{RNAP}$. At any given time $t$ and nuclei position $x$ along the AP axis, it was possible to calculate the probability for the promoter to be in the ON state (see Materials and methods and Appendix 1).

In this model, each kinetic parameter could be tuned independently to control the measured transcription dynamics features: Bcd binding rate constants ($k_i$, $k_b$) controlled the pattern boundary position, Bcd unbinding rate constants ($k_{-i}$) controlled the pattern steepness (*Estrada et al., 2016*; *Tran et al., 2018a*), the activation/deactivation rates ($k_{ON}$, $k_{OFF}$) controlled the fraction of active loci during steady state ($P_{Spot}$), and the RNAP firing rate ($\rho_{RNAP}$) controlled the mean loci intensity ($\mu_I$). To identify which processes were dependent on the number of Bcd BS, we first identified the parameters for the best fit of the model with the B6 data (*Figure 2E* and Appendix 2). Then, we allowed each of these parameters to vary, either alone or in combination, to fit the B9 (*Figure 2F* and *Figure 2—figure supplement 2*) and B12 data (*Figure 2G* and *Figure 2—figure supplement 2*). As they have more Bcd BS than B6, the fitting of the B9 and B12 data to the model also generated new parameters to account for higher Bcd-bound states (i.e. $k_{ON}(S_N)$ for N > 6). These simulations indicated that very good fits could be obtained for B9 and B12 by allowing only 3 of the $k_{-i}$ parameters to vary ($k_{-1}$, $k_{-2}$ & $k_{-6}$) (*Figure 2—figure supplement 2*) while the other parameters remained those identified for B6.

Given that the expression patterns of hb-P2 and all Bcd-only reporters reached a plateau in the anterior where Bcd concentration is likely in excess, we compared the activation rates $k_{ON}(S_N)$ of the promoter when $N$ = 6, 9 or 12 Bcd BS were occupied. Assuming that the number of bound Bcd proteins did not affect the switch OFF rate $k_{OFF}$, we found a fold change of ~4.5 between $k_{ON}(S_9)$ and $k_{ON}(S_6)$. This fold change is three times greater than the ratio of the Bcd BS numbers between B9 and B6. In contrast, there is almost no impact of adding three more sites when comparing B9 to B12 (even though it is the increase of ~1.33 times in the number of BS). This shows that the readout is not linear in the number of Bcd BS but that there is cooperativity/synergy between individual bound Bcd TF in the B9. Evidence for synergistic effects between several bound Bcd molecules is detailed in Appendix 3 and *Figure 2—figure supplement 3*.

## Hunchback reduces the burstiness of Bicoid-dependent transcription

Despite the same number of Bcd BS in the B6 and hb-P2 reporters, their expression pattern and dynamics were very different (*Figures 1 and 2A* and *Figure 2—figure supplement 1*). To determine whether this difference could be explained by the presence of BS for other TFs in the hb-P2 promoter, we used our synthetic approach to decipher the impact on the various features highlighted in our model when adding to the reporters BS for the two major partners of Bcd, Hb, and Zld also present in the hb-P2 promoter (*Figure 3A*). As our goal was to determine to which mechanistic step of our model each of these TF contributed, we purposefully started by adding BS in numbers that are much higher than in the hb-P2 promoter.

We first analyzed the impact of combining the Bcd BS with Hb BS. A H6 reporter containing only 6 Hb BS did not exhibit any MS2 signal from nc11 to nc13 (not shown). This indicated that the Hb protein alone, even with an abundance of Hb sites, could not activate transcription on its own. When combining 6 Hb BS with the 6 Bcd BS of B6 (henceforth named the H6B6 reporter, *Figure 3A*, *Video 5*), expression was detected in a similar domain to that of the B6 reporter, albeit with much higher fraction of active loci at any given time during the cycle (*Figure 3B*, middle panel). Across the embryo AP axis, the mean onset time of transcription after mitosis $T_0$ with the H6B6 reporter was not changed (p-values > 0.5) when compared to B6 (*Figure 3C*) and in the anterior region (excess of Bcd), the cumulative distribution of onset time $T_0$ was the same (*Figure 3D*). Interestingly, at their respective boundary positions where the Bcd concentration is limiting, the presence of Hb BS reduced to 325 ± 25 s the time required for the synthetic H6B6 reporter to reach the final decision to position its boundary (purple dashed line in *Figure 3E* and converging time, *Supplementary file 2*) when it was 425 ± 25 s for B6 (yellow dashed line in *Figure 3E* and converging time, *Supplementary file 2*). For H6B6, the fraction of nuclei with MS2 signal during the cycle exhibited a sigmoid-like pattern

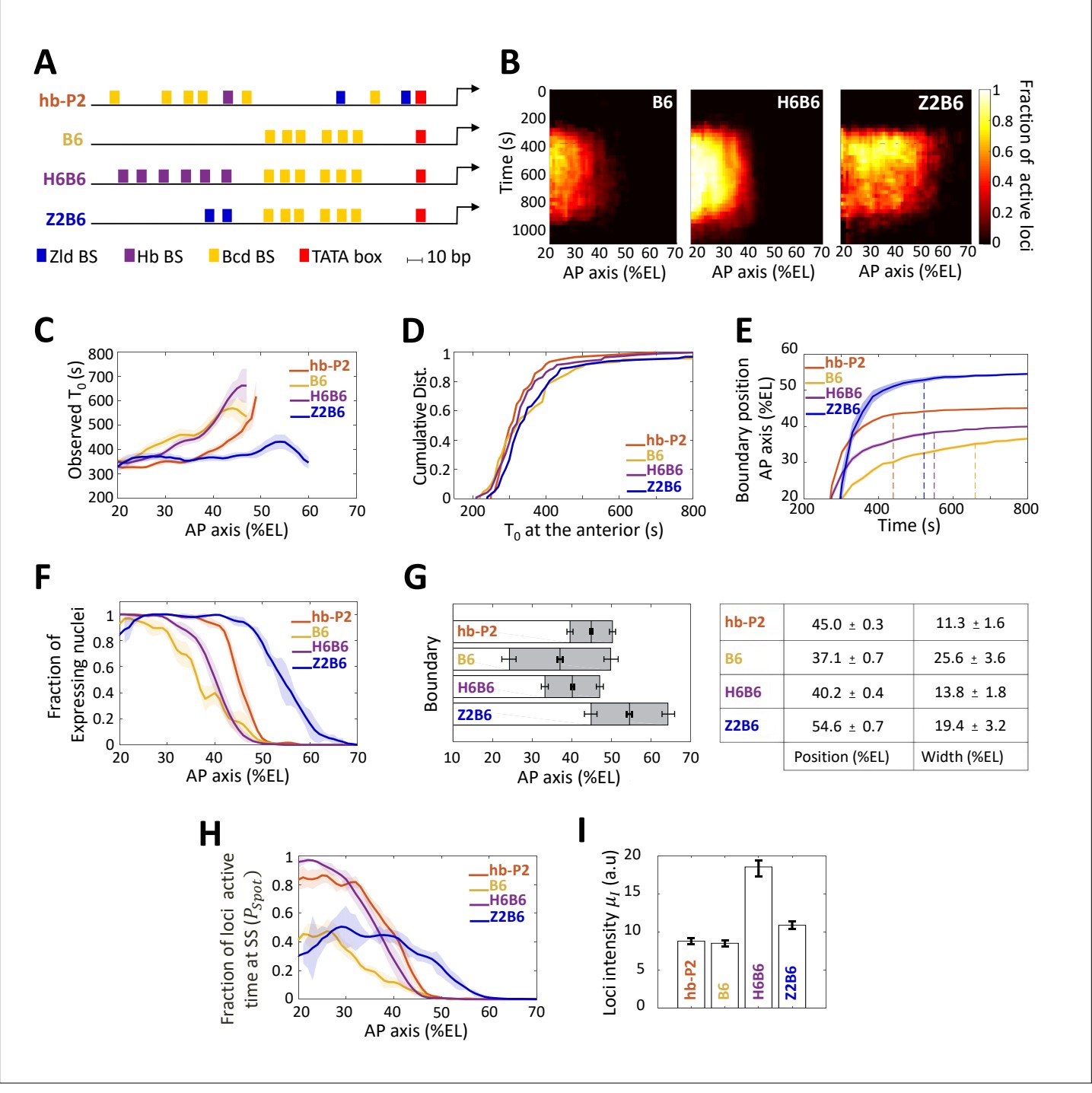

**Figure 3.** Transcription dynamics of the B6, H6B6, and Z2B6 reporters. (**A**) Arrangement of the binding sites for Bcd (yellow), Hb (purple), and Zld (blue) upstream of the TATA box (red) and the TSS (broken arrow) of each reporter. (**B**) Kymographs of mean fraction of active loci (colormap on the right) as a function of time (Y axis in s) and nuclei position along the AP axis (X axis in %EL) at nc13. (**C**) Mean time of first spot appearance $T_0$ (s) along the AP axis with shaded standard error of the mean and calculated only for loci with observed expression. (**D**) Cumulative distribution function of $T_0$ (s) at the anterior (20% ± 2.5%EL). (**E**) Boundary position (as %EL) of fraction of nuclei with MS2 signal along AP axis, with shaded 95% confidence interval, as a function of time. The dash vertical curves represent the time to reach the final decision boundary position ( ± 2 %EL). (**F**) Fraction of nuclei with any MS2 signal along the AP axis (%EL) with shaded standard error of the mean. (**G**) Boundary position and width were extracted by fitting the patterns fraction of expressing nuclei, (**F**) with a sigmoid function. Bar plots with 95% confidence interval for boundary position and width as the grey region placed symmetrically around the boundary position. Average values and confidence intervals are indicated in the adjacent table. (**H**) Fraction of loci active time ($P_{Spot}$) at steady state (time window of 600–800 s into nc13) as a function of nuclei position along AP axis. (**I**) Mean intensity ($\mu_I$) with standard error of

*Figure 3 continued on next page*

*Figure 3 continued*

active fluorescent loci detected in the anterior region (~20% ± 2.5% EL) at steady state. (**C–I**) reporter data are distinguished by color: hb-P2 (orange, n = 5 embryos), B6 (yellow, n = 5 embryos), H6B6 (purple, n = 7 embryos), and Z2B6 (blue, n = 3 embryos).

The online version of this article includes the following figure supplement(s) for figure 3:

**Figure supplement 1.** Transcription dynamics of the Z6 reporter (n = 2 embryos), B6 (n = 5 embryos) and Z2B6 (n = 3 embryos) expression patterns.

(*Figure 3F*) with, when compared to B6, a boundary slightly (only one nucleus length) shifted toward the posterior and a width reduced by half (*Figure 3G*). The kinetics of transcription regulation by the Hb protein was inferred from the fraction of the loci's active time ($P_{Spot}$) at steady state. In the anterior region, this fraction was always near saturation for the H6B6 reporter (~0.95–1) (*Figure 3H* and *Figure 2—figure supplement 1*), with very few nuclei exhibiting bursty expression. This non-bursty behavior of the H6B6 reporter contrasts with the highly bursty expression of B6 reporter (*Figure 2—figure supplement 1*). Meanwhile, in the anterior region, the mean fluorescence intensity of active H6B6 loci was at least twice higher than that of all synthetic Bcd-only reporters (*Figure 3I*).

To model H6B6 activity, the same formalism, as applied to B9 and B12 reporters, was used starting from the parameter values imposed from the fitted model of B6 and then varying those parameters, either alone or in combination, to fit the H6B6 data. The simulations indicated that a moderate fit to the data was obtained when varying only the $k_{ON}$ and $k_{OFF}$ parameters while varying in addition the 3 of the $k_{-i}$ parameters ($k_{-1}$, $k_{-2}$ & $k_{-6}$) allowed very good fitting of the model to the data (*Figure 2—figure supplement 2*).

Altogether, this suggests that Hb binding to the promoter accelerates the measurement of positional information by Bcd by improving both the unbinding kinetics of Bcd to its BS, which is consistent with the half reduction of the boundary steepness (*Figure 3G*) and the kinetics of activation/deactivation transcription rates, consistent with reduced burstiness (*Figure 3H* and *Figure 2—figure supplement 1*).

## Zelda lowers the Bcd threshold required for expression

As the hb-P2 promoter also contains Zld BS, we used our synthetic approach to investigate the role of Zld in the Bcd system. As a reporter with only 6 Zld BS (Z6) was strongly expressed along the whole AP axis (*Figure 3—figure supplement 1*), we had to reduce the number of Zld BS in our synthetic approach to analyze Zld effect. A Z2 reporter containing only 2 Zld BS did not exhibit any MS2 signal (not shown). The Z2B6 reporter (*Video 6*), combining 2 Zld BS with 6 Bcd BS (*Figure 3A*), exhibited a very different expression pattern when compared to B6 (*Figure 3B*, right panel). This expression pattern also varied with the nuclear cycles likely because of drastic changes in Zld transcriptional activity (*Figure 3—figure supplement 1*) rather than changes in the local concentration (*Dufourt et al., 2018*). For simplicity, we focused here on nc13. The onset time $T_0$ of the Z2B6 reporter was similar in the anterior to those of the B6, H6B6 and hb-P2 reporters (*Figure 3*, C-D) but unlike B6, H6B6 and hb-P2 it did not vary along the AP axis (*Figure 3C*). This suggests that Zld binding can

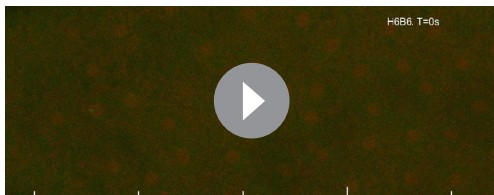

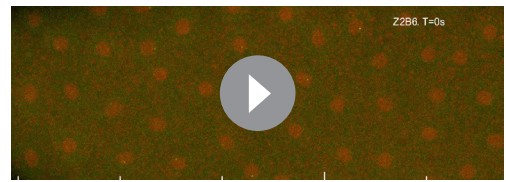

**Video 5.** Live imaging of transcription dynamics of H6B6 reporter. The videos have two channels: MCP-GFP channel (green) for monitoring the dynamics of nascent mRNA production and His-RFP (red) for nuclei detection. The capture frame is from 18% to 65% of embryo length. The anterior pole is on the left side of the frame. Position along the AP axis is indicated by white vertical bars positioned every 10% EL with the tallest one corresponding to 50% EL.

https://elifesciences.org/articles/74509/figures#video5

**Video 6.** Live imaging of transcription dynamics of Z2B6 reporter. The videos have two channels: MCP-GFP channel (green) for monitoring the dynamics of nascent mRNA production and His-RFP (red) for nuclei detection. The capture frame is from 20% to 70% of embryo length. The anterior pole is on the left side of the frame. Position along the AP axis is indicated by white vertical bars positioned every 10% EL with the tallest one corresponding to 50% EL.

https://elifesciences.org/articles/74509/figures#video6

accelerate Bcd-dependent transcription when Bcd is rate-limiting but has no effect when Bcd is in excess (*Figure 3C*). As observed with H6B6, the presence of Zld BS reduced to 300 ± 25 s the time required for the synthetic Z2B6 reporter to reach the final decision to position its boundary (blue dashed line in *Figure 3E* and converging time, *Supplementary file 2*) when it was 425 ± 25 s for B6 (yellow dashed line in *Figure 3E* and converging time, *Supplementary file 2*).

The most striking feature of the Z2B6 reporter was the drastic posterior shift of its expression boundary by ~17.5 %EL when compared to B6 (*Figure 3*, C-D). It indicates that the threshold of Bcd concentration required for activation is lowered when two Zld BS are present in the promoter together with 6 Bcd BS. Added to this, the pattern boundary width (*Figure 3G*) and in the anterior, both the active loci fraction $P_{Spot}$ (*Figure 3H* and *Figure 2—figure supplement 1*) and the loci intensity $\mu_I$ (*Figure 3I*), were very similar for the Z2B6 and B6 reporters. Therefore, we hypothesize that adding 2 Zld sites can accelerate and facilitate Bcd binding when Bcd is rate-limiting (i.e. increasing $k_b$ or $k_i$) without affecting the remaining parameters ($k_{-i}$, $k_{ON}$, $k_{OFF}$, $\rho_{RNAP}$). Consistent with this hypothesis, simulations for best fitting of the model to the data, starting from the parameters imposed by B6, indicate that a very good fit of the model to the Z2B6 data is obtained when only varying the Bcd binding rate $k_b$ (*Figure 2—figure supplement 2*).

Altogether, this suggests that Zld binding to the promoter accelerates the measurement of positional information by Bcd by facilitating Bcd binding when it is rate-limiting through an increase of the Bcd binding rate $k_b$, without affecting the kinetics of activation/deactivation transcription rates.

## A Bicoid-activity gradient with a short decay length

Since our synthetic Bcd-only reporters exclusively respond to Bcd, their expression boundary is exclusively dependent on specific thresholds of Bcd concentration, and this property was used to evaluate quantitatively the Bcd-activity gradient. For this, we reduced the amount of the Bcd protein by half in embryos from females, which were heterozygous for a CRISPR-induced deletion of the *bcd* gene ($\Delta$ *bcd*) (see Materials and methods). As the amount of Bcd protein is produced from each *bcd* allele independently of any other allele in the genome (*Liu et al., 2013*) and as changing the genetic dosage of *bcd* in the female leads to proportional changes in both mRNA and protein number in the embryo (*Petkova et al., 2014*), we assumed that embryos from wild-type females (2 X) express quantitatively twice as much Bcd proteins as embryos from $\Delta bcd/$ + females (1 X). In such Bcd-2X and Bcd-1X embryos, we compared the fraction of expressing nuclei along the AP axis as modeled at the top of *Figure 4A*. Data were obtained for B6 (*Figure 4B*) and B9 (*Figure 4C*). In addition, since Zld activity and concentration is homogeneous along the AP axis and likely independent of Bcd (as it is exclusively maternal), we also analyzed Z2B6 (*Figure 4D*) which provided useful information on how positional readout plays out at more posterior positions. For simplicity, we denoted $f_{2X}(x)$ the expression pattern in Bcd-2X embryos and $f_{1X}(x)$ the expression pattern in Bcd-1X embryos, with $x$ being the nuclei position along the AP axis.

To quantify the effects of perturbing the Bcd gradient, we first extracted from the experimental data the shift in position $\Delta(x)$ between two nuclei columns with the same expression distribution in Bcd-2X embryos ($f_{2X}(x)$) and in Bcd-1X embryos ($f_{1X}(x - \Delta(x))$), such that $f_{2X}(x) = f_{1X}(x - \Delta(x))$ (modeled in *Figure 4A*, middle panel). As all the expression patterns are noisy, we calculated the probability distribution of seeing a given shift $P(\Delta(x)|x)$ for each given position $x$ from our data and used a grey-scale log-probability map as a function of $x$ and $\Delta$ to present our results. An example of the log-probability map for the shift expected if the Bcd concentration was reduced by half at each position is shown at the bottom of *Figure 4A*. As expected, the prediction of $\Delta(x)$ should be most reliable in the boundary region (see Materials and methods and Appendix 4). From the combined log-probability map of the shift $\Delta(x)$ obtained from expression data of B6 (*Figure 4B*), B9 (*Figure 4C*) and Z2B6 (*Figure 4D*) reporters in Bcd-2X *vs* Bcd-1X embryos, we observed that the shift was very consistent in the zone between B6 and Z2B6's boundary regions (30% EL to 60% EL) (*Figure 4E*). Thus, it can be described by a constant value $\Delta(x) = \widetilde{\Delta}$, indicating that the Bcd activity gradient measured in this zone was exponential. From the data, the best fit value of $\widetilde{\Delta}$ was found to be 10.5% ± 1.0%EL (cyan dashed line in *Figure 4D*). Of note, the shift obtained at nc13 was larger than the shift obtained at nc12. However, the short length of nc12 (shorter than the time required for the Bcd-only reporters to reach steady state) likely introduces a bias in those measurements (*Figure 4—figure supplement 1*). Since the synthetic reporters, B6, B9 and Z2B6, are expected to position their boundary in Bcd-2X *vs*

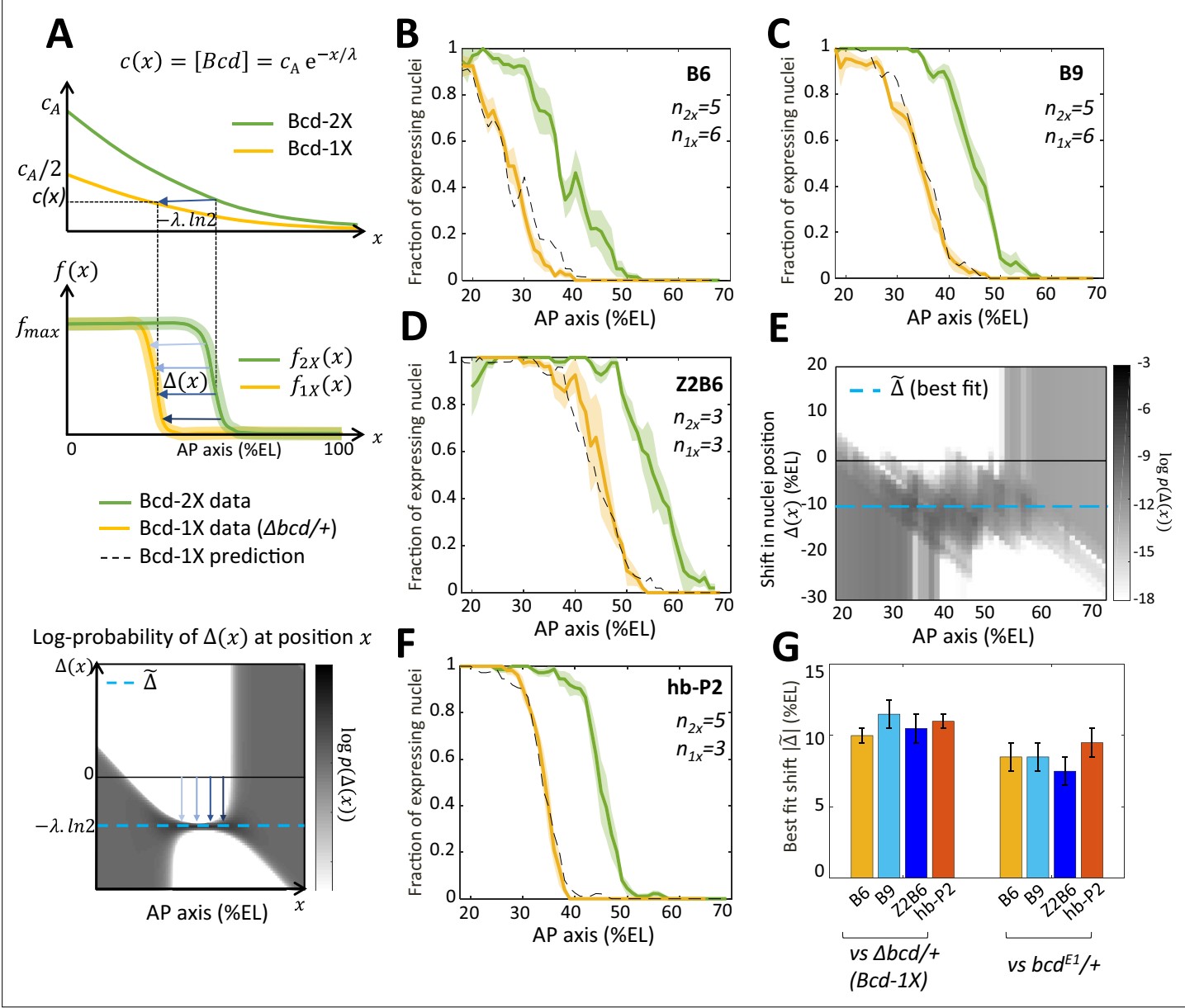

**Figure 4.** Bicoid thresholds measurements by the Bcd-only synthetic reporters. (**A**) Modeling the pattern shifts between Bcd-2X and Bcd-1X embryos. Top: The Bcd concentration gradient along the AP axis with its exponential decay length $\lambda$. At the anterior pole, Bcd concentration is $c_A$ in Bcd-2X embryos (solid green line) and $c_A/2$ in Bcd-1X embryos (solid yellow line). The distance between any two nuclei columns in Bcd-2X and Bcd-1X that have the same Bcd concentration (blue horizontal arrow) is given by $-\lambda \ln 2$. Middle: along the AP axis, expression pattern of a Bcd-dependent reporter in Bcd-2X embryos ($f_{2X}(x)$, solid green line) and in Bcd-1X embryos ($f_{1X}(x)$ solid yellow line). $\Delta(x)$: the shift in position (blue horizontal arrows) from a nuclei column in Bcd-2X embryos at position $x$ to one at Bcd-1X embryos with the same expression level, such that $f_{2X}(x) = f_{1X}(x - \Delta(x))$. Bottom: Cartoon of log-probability map of the shift $\Delta(x)$ based on the expression patterns in Bcd-2X and Bcd-1X (i.e. $f_{1X}(x)$ and $f_{2X}(x)$). Its value $\log p(\Delta(x))$ is represented on the grey scale. The blue vertical arrows denoting the shift correspond to the horizontal arrows with similar shade observed in the middle panel. If the Bcd gradient is the only source of positional information for the expression patterns, then the best fit value of $\Delta(x)$ given the probability map is $\widetilde{\Delta} = -\lambda \ln 2$ (horizontal blue dashed line). (**B–D and F**) Expression patterns of B6 (**B**), B9 (**C**), Z2B6 (**D**) and hb-P2 (**F**) reporters in embryos from wild-type (Bcd-2X, solid green lines with shaded errors) and $\Delta bcd/+$ (Bcd-1X, solid yellow lines with shaded errors) females. In each panel, the numbers of embryos for each construct and condition are also shown. Prediction of Bcd-1X patterns from the Bcd-2X patterns assuming a fitted constant shift (values in panel G) are shown as dashed black lines. (**E**) Log-probability map ($\log p(\Delta(x))$) of the shift $\Delta(x)$ (in %EL) at a given nuclei position in Bcd-2X embryos ($x$, in %EL), extracted from combined B6, B9, and Z2B6 reporters' data. The horizontal cyan dashed line represents the best fit value $\widetilde{\Delta}$ = 10.5 %EL from the log-probability map. (**G**) Comparison of the shift, with 95% confidence interval, in nuclei position

*Figure 4 continued on next page*

*Figure 4 continued*

from wild-type embryos to embryos from *Δbcd/* + females (left bars) and from wild-type embryos to embryos from *bcd^E1^/* + females (right bars) fitted individually to B6, B9, Z2B6 and hb-P2 reporters' data.

The online version of this article includes the following figure supplement(s) for figure 4:

**Figure supplement 1.** Comparison of fraction of expressing nuclei between nuclear cycles and between Bcd-2X and *bcd^E1^/* + flies.

Bcd-1X embryos at the same threshold of active Bcd concentration , the effective gradient highlighted by our analysis is exponential with an effective decay length $\lambda_{eff} = |\widetilde{\Delta}|/ln2 = 15\% \pm 1.4 \%EL$. We used the decay length for this effective gradient in the model to account for the pattern dynamics of B6, B9 and Z2B6 in Bcd-2X embryos and predict its pattern in Bcd-1X embryos. The predicted patterns from the model (black dashed curves) match well with the data (yellow curves) (*Figure 4B–D*). Lastly, the comparison of hb-P2 patterns in Bcd-2X *vs* Bcd-1X embryos indicated a shift of 11.0% ± 0.5%EL of the expression boundary (*Figure 4F*). As this value was indistinguishable from the shift $|\widetilde{\Delta}|$ obtained with data of the synthetic reporters above (*Figure 4*, B-D), we concluded that the measurement of positional information by the hb-P2 promoter is based entirely on the effective Bcd gradient with $\lambda_{eff}$ ~ 15% ± 1.4 %EL and does not involve input from other TF binding to the hb-P2 promoter.

Of note, the shift obtained for the hb-P2 MS2 reporter was significantly larger than the shift of 8% EL described for *hb* in previous studies using the *bcd^E1^* allele (*Houchmandzadeh et al., 2002*; *Porcher et al., 2010*). To understand this discrepancy, we measured the shift in the boundary positions of our hb-P2 and synthetic MS2 reporters in embryos from wild-type vs *bcd^E1^/* + females and confirmed that in this genetic background the shift of boundary position was 8% EL (*Figure 4G* and *Figure 4—figure supplement 1*, panel B-C). As the molecular lesion in the *bcd^E1^* allele introduces a premature stop codon downstream of the homeodomain (*Struhl et al., 1989*), these results suggest that the *bcd^E1^* allele likely allows the expression of a weakly functional truncated protein.

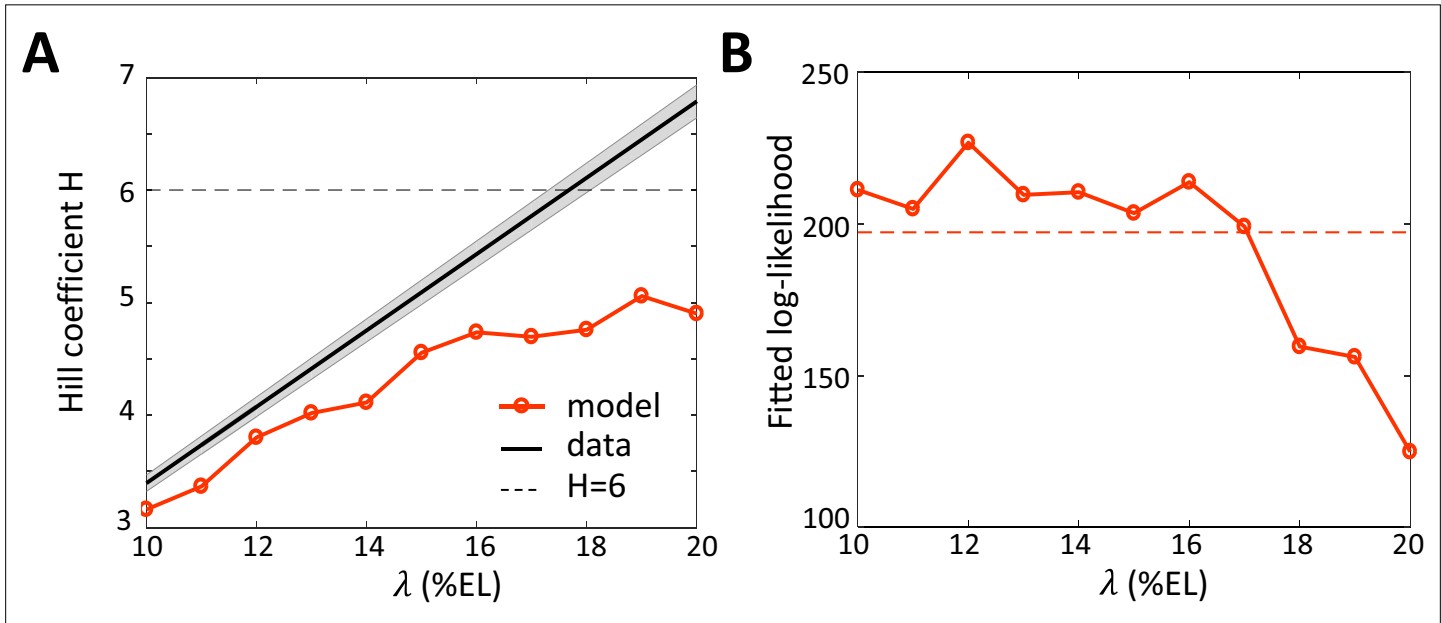

**Figure 5.** Fitting the data with models assuming different values for the Bcd gradient decay length $\boldsymbol{\lambda}$. (**A**) Hill coefficient $H$ (see Materials and methods) (solid red) in the steady state window (600 s-800s into nc13 interphase) calculated numerically from the best fitted models. Given that the pattern sharpness $\eta = H/\lambda$ is measured to be 0.34 from the hb-P2 data, the observed Hill coefficient as a function of $\lambda$ is given by the black line with shaded error. The physical limit of equilibrium sensing model with 6 BS ($H = 6$, black dashed line). (**B**) Log-likelihood of the best fitted models (solid red). The dashed line corresponds to the log-likelihood thresholds for a significantly worse fit (p-value = 0.05).

## The hb-P2 pattern steepness can be explained by an equilibrium model of concentration sensing

Assuming that nuclei extract positional information from an effective Bcd gradient with decay length $\lambda_{eff}$ ~ 15% EL, we reassessed the Hill coefficient (denoted as $H$), which reflects the cooperativity of *hb* regulation by Bcd (*Estrada et al., 2016*; *Gregor et al., 2007a*). For this, we fitted the pattern of expressing nuclei by the hb-P2 reporter (*Figure 1D*) to a sigmoid function. We transformed the fitted sigmoid function of position to a Hill function describing the transcription regulation function of hb-P2 by the Bcd protein concentration (see Materials and methods). Given the sigmoid function obtained from the data, the inferred Hill coefficient $H$ is proportional to assumed decay length $\lambda$ (black line in *Figure 5A*). Taking the observed effective decay length $\lambda_{eff}$ = 15% EL, we obtain $H$ ~ 5.2. As the hb-P2 promoter contains only six known Bcd BS, the value of $H$ = 5.2 for the Hill coefficient inferred assuming the decay length $\lambda_{eff}$ ~15% EL is now within the limit of concentration sensing with 6 Bcd BS ($H$ = 6, dashed horizontal line in *Figure 5A*) while the former value $H$ ~ 6.9 was not achievable without energy expenditure (*Estrada et al., 2016*) or positive feedback from Hb protein (*Lopes et al., 2011*).

To verify whether both the dynamics and sharpness of the hb-P2 expression pattern can be sufficiently explained by an equilibrium model of Bcd concentration sensing via $N$=6 Bcd BS, we fitted our model (*Figure 2*) to the kymograph of transcription dynamics by the hb-P2 reporter (*Figure 1C*). The effects of Hb and Zld are modeled implicitly by the kinetic rate constants. We varied the decay length $\lambda$ for the Bcd gradient varying from 10% to 20%EL (model assumptions in Materials and methods). The model assuming $\lambda = \lambda_{eff}$ = 15 %EL fitted the data significantly better (p-val <0.05) (*Figure 5B*) and reproduced a closer Hill coefficient at steady state (*Figure 5A*) than the model assuming $\lambda = \lambda_{detected}$ = 20 %EL.

Thus, lowering the decay length of the Bcd gradient to its effective value allows a more reliable fit of the model to the data and places back the Bcd system within the physical limits of an equilibrium model for concentration sensing.

## Discussion

Recently, synthetic approaches have been used to understand how the details of gene regulation emerge from the plethora of binding sites for transcription factors buried in genomes. In developmental systems, these approaches are starting to help us unravel the evolution of gene regulatory modules (reviewed in *Crocker and Ilsley, 2017*). In many cases, using high-throughput analysis of systematically mutagenized regulatory sequences, expression was measured through synthesis of easily detectable fluorescent proteins (*Farley et al., 2016*; *Gertz et al., 2009*; *Sharon et al., 2012*), RNA sequencing (*Melnikov et al., 2012*; *Patwardhan et al., 2012*) or antibody or FISH staining on fixed samples (*Erceg et al., 2014*; *Fuqua et al., 2020*). Even though these approaches allowed screening for a high number of mutated sequences with a very high resolution (single nucleotide level), the output measurements remained global and it was hard to capture the temporal dynamics of the transcription process itself. In addition, because effects of single mutations are frequently compensated by redundant sequences, it remained often difficult from these studies to highlight the mechanistic roles of the TF they bind to (*Vincent et al., 2016*). In this work, we combined the MS2 tagging system, which allows for a detailed measurement of the transcription process dynamics at high temporal resolution, with an orthogonal synthetic approach focusing on a few cis-regulatory elements with the aim of reconstructing from elementary blocks most features of *hb* regulation by Bcd. The number and placement of TF BS in our MS2 reporters are not identical to those found on the endogenous *hb* promoter and the number of combinations tested was very limited when compared to the high throughput approaches mentioned above. Nevertheless, this synthetic approach combined with quantitative analyses and modeling sheds light on the mechanistic steps of transcription dynamics (polymerase firing rate, bursting, licensing to be ON/OFF) involving each of the three TFs considered (Bcd, Hb, and Zld). Based on this knowledge from synthetic reporters and the known differences between them, we built an equilibrium model of transcription regulation which agrees with the data from the hb-P2 reporter expression.

Expression from the Bcd-only synthetic reporters indicate that increasing the number of Bcd BS from 6 to 9 shifts the transcription pattern boundary position toward the posterior region. This is expected as an array with more BS will be occupied faster with the required amount of Bcd molecules.

Increasing the number of Bcd BS from 6 to 9 also strongly increases the steepness of the boundary indicating that cooperativity of binding, or more explicitly a longer time to unbind as supported by our model fitting, is likely to be at work in this system. In contrast, adding three more BS to the 9 Bcd BS has very limited impact, indicating that either Bcd molecules bound to the more distal BS may be too far from the TSS to efficiently activate transcription or that the system is saturated with a binding site array occupied with 9 Bcd molecules. In the anterior with excess Bcd, the fraction of time when the loci are active at steady state also increases when adding 3 Bcd BS from B6 to B9. By assuming a model of transcription activation by Bcd proteins bound to target sites, the activation rate increases by much greater fold (~4.5 times) than the number of BS (1.5–2 times) suggesting a synergistic effect in transcription activation by Bcd.

The burstiness of the Bcd-only reporters in regions with saturating amounts of Bcd, led us to build a model in two steps. The first step of this model accounts for the binding/unbinding of Bcd molecules to the BS arrays. It is directly related to the positioning and the steepness of the expression boundary and thus to the measurement of positional information. The second step of this model accounts for the dialog between the bound Bcd molecules and the transcription machinery. It is directly related to the fluctuation of the MS2 signals including the number of firing RNAP at a given time (intensity of the signal) and bursting (frequency and length of the signal). Interestingly, while the first step of the process is achieved with an extreme precision (10% EL) (*Gregor et al., 2007a*; *Porcher et al., 2010*), the second step reflects the stochastic nature of transcription and is much noisier (*Desponds et al., 2016*; *Little et al., 2013*). Our model therefore also helps to understand and reconcile this apparent contradiction in the Bcd system.

As predicted by our original theoretical model (*Lucas et al., 2018*), 9 Bcd BS in a synthetic reporter appear sufficient to reproduce experimentally almost entirely the spatial features of the early *hb* expression pattern i.e. measurements of positional information. This is unexpected as the hb-P2 promoter is supposed to only carry 6 Bcd BS and leaves open the possibility that the number of Bcd BS in the *hb* promoter might be higher, as suggested previously (*Ling et al., 2019*; *Park et al., 2019*). Alternatively, it is also possible that even though containing 9 Bcd BS, the B9 reporter can only be bound simultaneously by less than 9 Bcd molecules. This possibility must be considered if for instance, the binding of a Bcd molecule to one site prevents by the binding of another Bcd molecule to another close by site (direct competition or steric hindrance). Even though we cannot exclude this possibility, we think that it is unlikely for several reasons: (*i*) some of the Bcd binding sites in the hb-P2 promoter are also very close to each other (see *Supplementary file 1*) and the design of the synthetic constructs was made by multimerizing a series of 3 Bcd binding sites with a similar spacing as found for the closest sites in the hb-P2 promoter (as shown in *Figure 1A* and *Supplementary file 1*); (*ii*) the binding of Bcd or other homeodomain containing proteins to two BS is generally increased by cooperativity when the sites are close to each other (as close as two base pairs for the paired homeodomain) compared to binding without cooperativity when they are separated by five base pairs or more (*Ma et al., 1996*; *Wilson et al., 1993*).

Importantly, even though we don't really know if the B9 and the hb-P2 promoter contain the same number of effective Bcd BS, the B9 reporter which solely contains Bcd BS recapitulates most spatial features of the hb-P2 reporter, clearly arguing that Bcd on its own brings most of the spatial (positional) information to the process. Interestingly, the B9 reporter is however much slower (2-fold) to reach the final boundary position than the hb-P2 reporter. This suggested that other maternally provided TFs binding to the hb-P2 promoter contribute to fast dynamics of the *hb* pattern establishment. Among these TFs, we focused on two known maternal partners of Bcd: Hb which acts in synergy with Bcd (*Porcher et al., 2010*; *Simpson-Brose et al., 1994*) and Zld, the major regulator of early zygotic transcription in fruit fly (*Liang et al., 2008*). Interestingly, adding Zld or Hb sites next to the Bcd BS array reduces the time for the pattern to reach steady state and modifies the promoter activity in different ways: binding of Zld facilitates the recruitment of Bcd at low concentration, making transcription more sensitive to Bcd and initiate faster while the binding of Hb affects strongly both the activation/deactivation kinetics of transcription (burstiness) and the RNAP firing rate. Thus, these two partners of Bcd contribute differently to Bcd-dependent transcription. Consistent with an activation process in two steps as proposed in our model, Zld will contribute to the first step favoring the precise and rapid measurements of positional information by Bcd without bringing itself positional information. Meanwhile, Hb will mostly act through the second step by increasing the level of transcription

through a reduction of its burstiness and an increase in the polymerase firing rate. Interestingly, both Hb and Zld binding to the Bcd-dependent promoter allow speeding-up the establishment of the boundary, a property that Bcd alone is not able to achieve. Of note, the hb-P2 and Z2B6 reporters contain the same number of BS for Bcd and Zld but they have also very different boundary positions and mean onset time of transcription $T_0$ following mitosis when Bcd is limiting. This is likely due to the fact that the two Zld BS in the hb-P2 promoter are not fully functional: one of the Zld BS is a weak BS while the other Zld BS has the sequence of a strong BS but is located too close from the TATA Box (5 bp) to provide full activity (*Ling et al., 2019*).

Zld functions as a pioneer factor by potentiating chromatin accessibility, transcription factor binding and gene expression of the targeted promoter (*Foo et al., 2014*; *Harrison et al., 2011*). Zld has recently been shown to bind nucleosomal DNA (*McDaniel et al., 2019*) and proposed to help establish or maintain cis-regulatory sequences in an open chromatin state ready for transcriptional activation (*Eck et al., 2020*; *Hannon et al., 2017*). In addition, Zld is distributed in nuclear hubs or microenvironments of high concentration (*Dufourt et al., 2018*; *Mir et al., 2018*). Interestingly, Bcd has been shown to be also distributed in hubs even at low concentration in the posterior of the embryo (*Mir et al., 2017*). These Bcd hubs are Zld-dependent (*Mir et al., 2017*) and harbor a high fraction of slow moving Bcd molecules, presumably bound to DNA (*Mir et al., 2018*). Both properties of Zld, binding to nucleosomal DNA and/or the capacity to form hubs with increased local concentration of TFs can contribute to reducing the time required for the promoter to be occupied by enough Bcd molecules for activation. In contrast to Zld, our knowledge on the mechanistic properties of the Hb protein in the transcription activation process is much more elusive. Hb synergizes with Bcd in the early embryo (*Simpson-Brose et al., 1994*) and the two TF contribute differently to the response with Bcd providing positional and Hb temporal information to the system (*Porcher et al., 2010*). Hb also contributes to the determination of neuronal identity later during development (*Hirono et al., 2017*). Interestingly, Hb is one of the first expressed members of a cascade of temporal TFs essential to determine the temporal identity of embryonic neurons in neural stem cells (neuroblasts) of the ventral nerve cord. In this system, the diversity of neuronal cell-types is determined by the combined activity of TFs specifying the temporal identity of the neuron and spatial patterning TFs, often homeotic proteins, specifying its segmental identity. How spatial and temporal transcription factors mechanistically cooperate for the expression of their target genes in this system is not known. Our work indicates that Hb is not able to activate transcription on its own but that it strongly increases RNAP firing probability and burst length of a locus licensed to be ON. Whether this capacity will be used in the ventral nerve cord and shared with other temporal TFs would be interesting to investigate.

The Bcd-only synthetic reporters also provided an opportunity to scrutinize the effect of Bcd concentration on the positioning of the expression domain boundaries. This question has been investigated with endogenous *hb* in the past, always giving a smaller shift than expected given the decay length of 20% EL for the Bcd protein gradient (*Bergmann et al., 2007*; *Liu et al., 2013*; *Porcher et al., 2010*) and arguing against the possibility that positional information in this system could solely be dependent on Bcd concentration. When comparing the transcription patterns of the B9 reporter in Bcd-2X flies and Bcd-1X flies, we detected a shift of ~10.5 ± 1% EL of the boundary position. This shift revealed a gradient of Bcd activity with an exponential decay length of ~15 ± 1.4% EL (~75 μm), significantly smaller than the value observed directly (20% EL, ~ 100 μm) with immuno-staining for the Bcd protein gradient (*Houchmandzadeh et al., 2002*) but closer from the value of 16.4% EL obtained with immuno-staining for Bcd of the Bcd-GFP gradient (*Liu et al., 2013*). Given the discrepancies of previous studies concerning the measurements of the Bcd protein gradient decay length (see Appendix 5 and *Supplementary file 3*), our work calls for a better quantification to determine how close the decay length of the Bcd protein gradient is from the decay length of the Bcd activity gradient uncovered here. Our work opens the possibility that the effective decay length of 15% EL corresponds to a population of 'active' or 'effective' Bcd distributed in steeper gradient than the Bcd protein gradient observed by immunodetection which would include all Bcd molecules. Bcd molecules have been shown to be heterogenous in intranuclear motility, age and spatial distributions but to date, we do not know which population of Bcd can access the target gene and activate transcription (*Tran et al., 2020*). The existence of two (or more) Bicoid populations with different mobilities (*Abu-Arish et al., 2010*; *Fradin, 2017*; *Mir et al., 2018*) obviously raises the question of the underlying gradient for each of them. Also, the dense Bcd hubs persist even in the posterior region where the

Bcd concentration is low (*Mir et al., 2017*). As the total Bcd concentration decreases along the AP axis, these hubs accumulate Bcd with increasing proportion in the posterior, resulting in a steeper gradient of free-diffusing Bcd molecules outside the hubs. At last, the gradient of newly translated Bcd was also found to be steeper than the global gradient (*Durrieu et al., 2018*). Finally and most importantly, reducing by half the Bcd concentration in the embryo induced a similar shift in the position of the hb-P2 reporter boundary as that of the Bcd-only reporters. This further argues that this gradient of Bcd activity is the principal and direct source of positional information for *hb* expression.

The effective Bcd gradient found here rekindles the debate on how a steep *hb* pattern can be formed in the early nuclear cycles. With the previous value of $\lambda = 20\%$ EL for the decay length of the Bcd protein gradient (*Houchmandzadeh et al., 2002*), the Hill coefficient inferred from the fraction of loci's active time at steady state $P_{Spot}$ is ~6.9, beyond the theoretical limit of the equilibrium model of Bcd interacting with six target BS of the *hb* promoter (*Estrada et al., 2016*; *Hopfield, 1974*). This led to hypotheses of energy expenditure in Bcd binding and unbinding to the sites (*Estrada et al., 2016*), out-of-equilibrium transcription activation (*Desponds et al., 2020*), *hb* promoters containing more than 6 Bcd sites (*Lucas et al., 2018*; *Park et al., 2019*) or additional sources of positional information (*Tran et al., 2018a*) to overcome this limit. The effective decay length $\lambda_{eff}$ ~15% EL, found here with a Bcd-only reporter but also hb-P2, corresponds to a Hill coefficient of ~5.2, just below the physical limit of an equilibrium model of concentration sensing with 6 Bcd BS alone. Of note, a smaller decay length also means that the effective Bcd concentration decreases faster along the AP axis. In the Berg & Purcell limit (*Berg and Purcell, 1977*), the time length to achieve the measurement error of 10% at hb-P2 expression boundary with $\lambda$=15% EL is ~2.1 times longer than with $\lambda$=20% EL (see Appendix 6 where we show the same argument holds regardless of estimated parameter values). This points again to the trade-off between reproducibility and steepness of the *hb* expression pattern, as described in *Tran et al., 2018a* and reinforces the importance of Hb and Zelda in speeding-up the process.

## Materials and methods

### *Drosophila* stocks

Embryos were obtained from crosses between males carrying MS2 reporters and females carrying the maternally expressed MCP-NoNLS-eGFP (*Garcia et al., 2013*) and His2Av-mRFP (Bloomington # 23561) transgenes both on the second chromosome. Embryos with reduced activity of Bcd were obtained from females which were in addition heterozygotes for the *bcdΔ* molecular null allele or for the *bcd^E1* allele (*bcd^6*). Unless otherwise specified, all MS2 reporters were inserted at the vk33 docking site (Bloomington # 9750) via $\phi$C31-mediated integration system (*Venken et al., 2006*) by BestGene. The site of insertion was chosen because the transcription dynamics of the original hb-P2 reporter (*Lucas et al., 2018*) inserted at this site was indistinguishable from the transcription dynamics of two randomly inserted siblings (*Figure 1—figure supplement 1*, panels A-C). All fly stocks were maintained at 25°C.

### MS2 reporters

The hb-P2 MS2 reporter was obtained by cloning the 745 bp (300 bp upstream of the transcription start site to 445 bp downstream, including the hb intron) located just upstream start codon of Hunchback protein (*Drosophila melanogaster*) from the previously used hb-MS2ΔZelda reporter (*Lucas et al., 2018*) into the attB-P[acman]-Cm^R-BW plasmid. The synthetic MS2 reporters were created by replacing the *hb* region in hb-P2 MS2 by the *hsp70Bb* promoter and the synthetic sequences containing specific combinations of binding sites. GGGATTA was used as a Bcd binding site, CAGGTAG as a Zld binding site and TCAAAAAATAT or TCAAAAAACTAT as Hb binding sites. The sequences of these promoters are given in the *Supplementary file 1*.

### Generation of the *Δbcd* mutant by CRISPR

The *Δbcd* molecular null allele was generated by CRISPR/Cas9 genome editing using the scarless strategy described in *Gratz et al., 2015*. gRNA sequences were designed to induce Cas-9-dependent double strand DNA hydrolysis 460 pb upstream of the *bcd* gene TATA box and 890 bp downstream of the Bcd stop codon. For this, double stranded oligonucleotides (sequences available in *Supplementary file 1*: Oligo for 5' cut Fw, Oligo for 5' cut Rv, Oligo for 3' cut Fw, Oligo for 3' cut Rv) were

inserted into the Bbs-1 restriction site of pCFD3-dU6_3gRNA vector (*Port et al., 2014*). The two homology arms flanking the cleavage sites were amplified from genomic DNA by PCR using the NEB Q5 high fidelity enzyme and specific oligonucleotides (sequences available in *Supplementary file 1*: Bcdnull_5 HR_fw, Bcdnull_5 HR_rv, Bcdnull_3 HR_fw, Bcdnull_3 HR_rv). The scarless-DsRed sequence was amplified with Q5 from the pHD-ScarlessDsRed vector using specific oligonucleotides (sequences available in *Supplementary file 1*: Bcdnull_DsRed_fw, Bcdnull_DsRed_rv). The three PCR amplified fragments were mixed in equimolar ratio with the 2835 bp SapI-AarI fragment of pHD-ScarlessDsRed for Gibson assembly using the NEBuilder system. Injections and recombinant selection based on DsRed expression in the eye were performed by BestGene. For transformants, sequences at the junctions between deletion break point and the inserted dsRed marker were amplified by PCR and verified by sequencing.

## Live embryo imaging

Sample preparation and live imaging of transcription was performed as in *Perez-Romero et al., 2018*. Briefly, embryos were collected 30 min after egg laying, dechorionated by hand and mounted on coverslips covered in heptane-dissolved glue and immersed in 10 S Voltatef oil (VWR). All embryos were imaged between nc10 and nc14 at stable temperature (25 °C) on a LSM780 confocal microscope equipped with a 40 x (1.4 NA Plan-Apochromat) oil immersion objective. For each embryo, a stack of images is acquired (0.197 μm pixel size, 8 bit per pixel, 0.55μs pixel dwell time, confocal pinhole diameter of 92 μm, 0.75 μm distance between consecutive images in the stack, ~1200 × 400 pixels image size). GFP and RFP proteins were imaged with 488 nm and 561 nm lasers, respectively, with appropriate power output. Embryo size and position of the imaged portion is calculated through imaging and measurement of a tiled image of the sagittal plane of the embryo.

## Data extraction

Data extraction from MS2 Videos was performed as in *Lucas et al., 2018* using the LiveFly toolbox (*Tran et al., 2018b*). In brief, nuclei were segmented in a semi-automatic manner based on His2Av-mRFP channel. The active MS2 loci detection was performed in 3D using a thresholding method. The pixel values at the detected loci location were then fitted with a gaussian kernel to obtain the MS2 loci intensity. The expression data containing each nucleus' position along AP axis and MS2 loci intensity trace over time was exported.

From each time trace in individual nuclei at nc13 (see an example in panel A of *Figure 2—figure supplement 1*), we extracted three features: the detection of MS2 expression during the nuclear interphase (taking only 0 or 1 as values), $T_0$ the onset time of transcription detection following mitosis and $P_{Spot}$ the probability of detecting a spot during the steady state window. When calculating $T_0$, to account for the mitotic waves (i.e. nuclei at the anterior poles may divide first leading to uneven timings of chromatin decompaction or reentrance of Bcd into the nuclei along the AP axis) (*Vergassola et al., 2018*), we first defined the 'birth moment' of each nucleus as the time when the segregation from its sibling after mitosis is complete (see (*Lucas et al., 2018*) for the detailed procedure). The intensity trace in each nucleus was then trimmed so as it starts at its respective nucleus' birth time and $T_0$ measured as the time of the first MS2 spot appearance in regard to this birth time. The steady state window for $P_{Spot}$ was defined to be 600–800 s into nc13 due to a transient 'surge' in transcription activity with the hb-P2 reporter inserted in the vk33 (*Figure 1—figure supplement 1*).

## Quantifying pattern sharpness, boundary width and Hill coefficient

To quantify the sharpness of the transcription patterns of MS2 reporters, we fit the patterns along the AP axis to a sigmoid function:

$$f_{Sigmoid}\left(x\right) = f_{max}\frac{e^{-x\eta}}{e^{-x\eta}+e^{-x_0\eta}}. \tag{1}$$

In *Equation 1*, $x$ is the position of the nuclei. $f_{max}$ is the maximum expression level at the anterior pole ($f_{max} = f_{Sigmoid}\left(x = -\infty\right)$). $x_0$ is the expression boundary position ($f_{Sigmoid}\left(x_0\right) = f_{max}/2$). $\eta$ is the scaling coefficient of the AP axis. $\eta$ also corresponds to the pattern sharpness as it is the derivative of the sigmoid function at the boundary position divided by $f_{max}$.

From the fitted pattern, we define the boundary width as the distance between two nuclei columns with $f_{Sigmoid}\left(x\right)$ of 5% and 95% of maximum expression level $f_{max}$.

$$\text{Boundary width} = 2\ln\left(\frac{0.95}{0.05}\right)/\eta = 5.59/\eta. \tag{2}$$

We assume an exponential Bcd gradient with the decay length $\lambda$ ($[Bcd] = c_0 e^{-x/\lambda}$), where $c_0$ is Bcd concentration at $x=0$. We replace $x = -\lambda \log\left([Bcd]/c_0\right)$ and find the gene expression pattern from the promoter:

$$
\begin{aligned}
f_{regulation}\left([Bcd]\right) &= f_{max}\frac{e^{-\lambda.\log([Bcd]/c_0)\eta}}{e^{-\lambda.\log([Bcd]/c_0)\eta}+e^{-x_0\eta}} \\
&= f_{max}\frac{\left(\frac{[Bcd]}{c_0}\right)^{\lambda\eta}}{\left(\frac{[Bcd]}{c_0}\right)^{\lambda\eta}+e^{-x_0\eta}} = f_{max}\frac{[Bcd]^{\lambda\eta}}{[Bcd]^{\lambda\eta}+c_0^{\lambda\eta}e^{-x_0\eta}}
\end{aligned}
\tag{3}
$$

One should note that in *Equation 3* the Hill function with the Hill coefficient $H$ depends on both the pattern sharpness $\eta$ and the decay length of the Bcd gradient $\lambda$:

$$H = \lambda\eta. \tag{4}$$

In *Equation 4*, the pattern sharpness $\eta$ can be extracted directly from the MS2 Videos. Therefore, the assumptions on the decay length $\lambda$ will determine the inferred Hill coefficient $H$ and consequently the requirements of Bcd binding cooperativity and energy expenditure to achieve such coefficients (*Estrada et al., 2016*; *Tran et al., 2018a*).

## Simulating the model of transcription regulation by Bcd

For a model of transcription regulation with $N$ Bcd binding sites, the system can be in 2xN +2 states, that consists of $(N + 1)$ binding array states ($S_0$ to $S_N$) and two transcriptional states (*ON* and *OFF*), as described in detail in the Appendix 1.

## Calculating the shift in pattern along AP Axis

We quantify the shift of the MS2 expression patterns along the AP axis from Bcd-2x to Bcd-1x flies in terms of the probability distribution of the shift $\Delta\left(x\right)$ from position $x$ position $P_{R|X}\left(R|X = x\right)$ given by:

$$P_{\Delta(x)}\left(\Delta|x\right) = P_{R|X}\left(R = x + \Delta|x\right). \tag{5}$$

Given this probability distribution $P_{\Delta(x)}\left(\Delta\right)$ as a function of $x$, we can find a constant value of the shift $\Delta\left(x\right) = \widetilde{\Delta}$ that best describes the observed shift for all positions $x$ within $\epsilon X$:

$$\widetilde{\Delta} = \arg\max_{\Delta}\int_{\epsilon X} p\left(\Delta\left(x\right) = \Delta\right) dx, \tag{6}$$

as described in detail in the Appendix 4.

## Fitting the models of transcription regulation by Bcd

We fit the models of Bcd binding/unbinding to binding sites and activation of transcription, each with a different value of the Bcd gradient decay length $\lambda$, to the transcription dynamics by the synthetic reporters (*Figure 2* and *Figure 2—figure supplement 2*) and hb-P2 reporter (*Figure 5*) as described in detail in the Appendix 2.

## Acknowledgements

We thank A Coulon, A Ramaekers, A Taddei and the members of the Nuclear Dynamics Unit for fruitful discussions. We are indebted to BestGene Inc for transgenics and genome edition and to Patricia Le Baccon and the Imaging Facility PICT-IBiSA of the Institut Curie. This work is supported by PSL IDEX REFLEX Grant for Mesoscopic Biology (ND, AMW, MC), ANR-19-CE13-0025 FIREFLY (ND, AMW, CF), ARC PJA20191209543 (ND), ARC PJA20151203341 (ND) and ANR-11-LABX-0044 DEEP Labex (ND, HT), Marie Skłodowska-Curie grant agreement No 666,003 (GF), ARC DOC42021020003330 (GF) and NSERC discovery grant RGPIN-2015–06362 (CF, CPR). The funders had no role in study design, data collection and analysis, decision to publish, or preparation of the manuscript.

# Additional information

### Competing interests

Aleksandra M Walczak: Senior editor, eLife. The other authors declare that no competing interests exist.

### Funding

| Funder | Grant reference number | Author |
|---|---|---|
| H2020 Marie Skłodowska-Curie Actions | 666003 | Gonçalo Fernandes |
| Agence Nationale de la Recherche | PSL IDEX Mesoscopic Biology | Nathalie Dostatni Aleksandra M Walczak Mathieu Coppey |
| Agence Nationale de la Recherche | ANR-19-CE13-0025 | Nathalie Dostatni Aleksandra M Walczak Cecile Fradin |
| Agence Nationale de la Recherche | ANR-11-LABX-0044 DEEP Labex | Nathalie Dostatni Huy Tran |
| Fondation ARC pour la Recherche sur le Cancer | DOC42021020003330 | Gonçalo Fernandes |
| Fondation ARC pour la Recherche sur le Cancer | PJA20191209543 | Nathalie Dostatni |
| Natural Sciences and Engineering Research Council of Canada | RGPIN-2015-06362 | Carmina Perez Romero Cecile Fradin |
| Fondation ARC pour la Recherche sur le Cancer | PJA20151203341 | Nathalie Dostatni |
| Institut Curie | | Huy Tran |

The funders had no role in study design, data collection and interpretation, or the decision to submit the work for publication.

### Author contributions

Gonçalo Fernandes, Data curation, Formal analysis, Investigation, Methodology, Resources, Validation, Visualization, Writing – review and editing; Huy Tran, Data curation, Formal analysis, Investigation, Methodology, Resources, Software, Validation, Visualization, Writing – original draft, Writing – review and editing; Maxime Andrieu, Conceptualization, Investigation, Methodology, Resources; Youssoupha Diaw, Investigation; Carmina Perez Romero, Conceptualization, Methodology, Resources; Cécile Fradin, Conceptualization, Funding acquisition, Project administration, Supervision; Mathieu Coppey, Conceptualization, Formal analysis, Funding acquisition, Project administration, Software, Supervision, Validation, Visualization, Writing – review and editing; Aleksandra M Walczak, Conceptualization, Funding acquisition, Methodology, Project administration, Supervision, Validation, Visualization, Writing – review and editing; Nathalie Dostatni, Conceptualization, Formal analysis, Funding acquisition, Methodology, Project administration, Supervision, Validation, Visualization, Writing – review and editing

### Author ORCIDs

Gonçalo Fernandes http://orcid.org/0000-0001-8352-2581
Huy Tran http://orcid.org/0000-0002-9057-060X
Mathieu Coppey http://orcid.org/0000-0001-8924-3233
Aleksandra M Walczak http://orcid.org/0000-0002-2686-5702
Nathalie Dostatni http://orcid.org/0000-0002-1562-6167

### Decision letter and Author response

Decision letter https://doi.org/10.7554/eLife.74509.sa1
Author response https://doi.org/10.7554/eLife.74509.sa2

## Additional files

### Supplementary files

• Supplementary file 1. Promoter sequences of hb-P2, synthetic MS2 reporters and oligonucleotides required for generating the Δ bcd molecular null allele. The sequences are shown in modules, as arranged in *Figure 1* in the main text. The TATA boxes of hb-P2 and HSBG promoter are highlighted in bold. In the binding array sequences, the binding sites for each protein are highlighted in grey (Bcd), green (Hb) and yellow (Zelda). For clarity, the binding array sequences are shown up to the TATA box of HSBG promoter.

• Supplementary file 2. Position and width of the gene expression boundary based on fraction of expression nuclei feature (*Figures 1H and 3G*) and fraction of active loci $P_{spot}$ feature (*Figures 2A and 3H*) in the steady window (600–800 s into nc13) for hb-P2 and synthetic reporters in Bcd-2X, shown with 95% confidence interval. For the fraction of expression nuclei, also shown is the time to reach the final activation decision boundary ( ± 2 %EL) starting from the detection the first spot (~225 s) after mitosis.

• Supplementary file 3. Estimated values of gradient decay length for the Bcd protein and Bcd fluorescently tagged protein gradients from previous works.

• Supplementary file 4. Estimated values of Bcd concentration at the anterior ($c_A$), diffusion coefficient ($D$) and the size of the target for binding ($a$) from previous work.

• Supplementary file 5. Bcd search time for the target site ($t_{bind}$) and the readout time ($T$) required for 10% error in Bcd concentration readout at hb-P2 boundary position (–4.9 %EL), calculated for different values of Bcd concentration at the anterior ($c_A$), diffusion coefficient ($D$) and targets size ($a$) in the Berg & Purcell limit (*Berg and Purcell, 1977*).

• Supplementary file 6. Expected fold change in the activation rates in the anterior region (saturating Bcd concentration) $k_{ON} \left( S_{i=N} \right)$ between B9 and B6 and between B12 and B6. The fold changes are shown for different schemes of activation and values of $K$ (*Figure 2—figure supplement 3*). The fold changes above the value calculated from the data (~4.5) are made bold.

• Transparent reporting form

### Data availability

All the movies used are deposited at Zenodo and are accessible through a community link: https://zenodo.org/communities/hb-synthetic/. Each dataset (several movies of embryos with the same genotype) are referenceable and can be accessed through their individual DOI: hb-P2: https://doi.org/10.5281/zenodo.5361599 B6: https://doi.org/10.5281/zenodo.5457893 B9: https://doi.org/10.5281/zenodo.5457944 B12: https://doi.org/10.5281/zenodo.5458309 H6: https://doi.org/10.5281/zenodo.5459332 H6B6: https://doi.org/10.5281/zenodo.5458777 Z6: https://doi.org/10.5281/zenodo.5459338 Z2: https://doi.org/10.5281/zenodo.5459336 Z2B6: https://doi.org/10.5281/zenodo.5459314 bcd1X(delta)-hb-P2: https://doi.org/10.5281/zenodo.5463611 bcd1X(delta)-B6: https://doi.org/10.5281/zenodo.5994754 bcd1X(delta)-B9: https://doi.org/10.5281/zenodo.5463618 bcd1X(delta)-Z2B6: https://doi.org/10.5281/zenodo.5994806 bcd1X(bcdE1)-hb-P2: https://doi.org/10.5281/zenodo.5464256 bcd1X(bcdE1)-B6: https://doi.org/10.5281/zenodo.5464655 bcd1X(bcdE1)-B9: https://doi.org/10.5281/zenodo.5465647 bcd1X(bcdE1)-B12: https://doi.org/10.5281/zenodo.5466741 bcd1X(bcdE1)-H6B6: https://doi.org/10.5281/zenodo.5466785 bcd1X(bcdE1)-Z2B6: https://doi.org/10.5281/zenodo.5466823 B6-hbpromoter: https://doi.org/10.5281/zenodo.5473374 hb-P2-II: https://doi.org/10.5281/zenodo.5477862 hb-P2-III: https://doi.org/10.5281/zenodo.5477926. https://doi.org/10.5281/zenodo.5459332.

The following datasets were generated:

| Author(s) | Year | Dataset title | Dataset URL | Database and Identifier |
|---|---|---|---|---|
| Fernandes G, Tran H | 2021 | hb-P2 MS2 reporter data | https://doi.org/10.5281/zenodo.5361599 | Zenodo, 10.5281/zenodo.5361599 |
| Fernandes G, Tran H | 2021 | B6 MS2 reporter data | https://doi.org/10.5281/zenodo.5457893 | Zenodo, 10.5281/zenodo.5457893 |

*Continued on next page*

*Continued*

| Author(s) | Year | Dataset title | Dataset URL | Database and Identifier |
|---|---|---|---|---|
| Fernandes G, Tran H | 2021 | B9 MS2 reporter data | https://doi.org/10.5281/zenodo.5457944 | Zenodo, 10.5281/zenodo.5457944 |
| Fernandes G, Tran H | 2021 | B12 MS2 reporter data | https://doi.org/10.5281/zenodo.5458309 | Zenodo, 10.5281/zenodo.5458309 |
| Fernandes G, Tran H | 2021 | H6 MS2 reporter data | https://doi.org/10.5281/zenodo.5459332 | Zenodo, 10.5281/zenodo.5459332 |
| Fernandes G, Tran H | 2021 | H6B6 MS2 reporter data | https://doi.org/10.5281/zenodo.5458777 | Zenodo, 10.5281/zenodo.5458777 |
| Fernandes G, Tran H | 2021 | Z6 MS2 reporter data | https://doi.org/10.5281/zenodo.5459338 | Zenodo, 10.5281/zenodo.5459338 |
| Fernandes G, Tran H | 2021 | Z2 MS2 reporter data | https://doi.org/10.5281/zenodo.5459336 | Zenodo, 10.5281/zenodo.5459336 |
| Fernandes G, Tran H | 2021 | Z2B6 MS2 reporter data | https://doi.org/10.5281/zenodo.5459314 | Zenodo, 10.5281/zenodo.5459314 |
| Fernandes G, Tran H | 2021 | hb-P2 MS2 reporter in Bcd-1X (delta-bcd) data | https://doi.org/10.5281/zenodo.5463611 | Zenodo, 10.5281/zenodo.5463611 |
| Fernandes G, Tran H | 2022 | B6 MS2 reporter in Bcd-1X (delta-bcd) data | https://doi.org/10.5281/zenodo.5994754 | Zenodo, 10.5281/zenodo.5994754 |
| Fernandes G, Tran H | 2021 | B9 MS2 reporter in Bcd-1X (delta-bcd) data | https://doi.org/10.5281/zenodo.5463618 | Zenodo, 10.5281/zenodo.5463618 |
| Fernandes G, Tran H | 2022 | Z2B6 MS2 reporter in Bcd-1X (delta-bcd) data | https://doi.org/10.5281/zenodo.5994806 | Zenodo, 10.5281/zenodo.5994806 |
| Fernandes G, Tran H | 2021 | hb-P2 MS2 reporter in Bcd-1X (bcdE1) data | https://doi.org/10.5281/zenodo.5464256 | Zenodo, 10.5281/zenodo.5464256 |
| Fernandes G, Tran H | 2021 | B6 MS2 reporter in Bcd-1X (bcdE1) data | https://doi.org/10.5281/zenodo.5464655 | Zenodo, 10.5281/zenodo.5464655 |
| Fernandes G, Tran H | 2021 | B9 MS2 reporter in Bcd-1X (bcdE1) data | https://doi.org/10.5281/zenodo.5465647 | Zenodo, 10.5281/zenodo.5465647 |
| Fernandes G, Tran H | 2021 | B12 MS2 reporter in Bcd-1X (bcdE1) data | https://doi.org/10.5281/zenodo.5466741 | Zenodo, 10.5281/zenodo.5466741 |
| Fernandes G, Tran H | 2021 | H6B6 MS2 reporter in Bcd-1X (bcdE1) data | https://doi.org/10.5281/zenodo.5466785 | Zenodo, 10.5281/zenodo.5466785 |
| Fernandes G, Tran H | 2021 | Z2B6 MS2 reporter in Bcd-1X (bcdE1) data | https://doi.org/10.5281/zenodo.5466823 | Zenodo, 10.5281/zenodo.5466823 |
| Fernandes G, Tran H | 2021 | B6-hbpromoter MS2 reporter data | https://doi.org/10.5281/zenodo.5473374 | Zenodo, 10.5281/zenodo.5473374 |
| Fernandes G, Tran H | 2021 | hb-P2-II MS2 reporter data | https://doi.org/10.5281/zenodo.5477862 | Zenodo, 10.5281/zenodo.5477862 |
| Fernandes G, Tran H | 2021 | hb-P2-III MS2 reporter data | https://doi.org/10.5281/zenodo.5477926 | Zenodo, 10.5281/zenodo.5477926 |

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

## Appendix 1

## Simulating the model of transcription regulation by Bcd

For a model of transcription regulation with $N$ Bcd binding sites, the system can be in 2x$N$ +2 states, that consists of $(N + 1)$ binding array states ($S_0$ to $S_N$) and two transcriptional states ($ON$ and $OFF$).

We define the state probability vector $\bar{u} = \left[u_1, \ u_2, \ \ldots u_{2*(N+1)}\right]^T$ ($\Sigma_i u_i = 1$), where:

$$u_i = \begin{bmatrix} Probability \ \left(S_{i-1}, OFF\right) \ \text{with } i \leq N + 1 \\ Probability \ \left(S_{i-1}, ON\right) \ \text{with } i > N + 1. \end{bmatrix} \tag{A1-1}$$

At $t = 0$, we assume that all the Bcd binding sites are free, and the promoter is OFF. Therefore, we have the initial condition: $\bar{u}_0 = \left[u_1 = 1, u_{i>1} = 0\right]^T$.

As all the transitions between promoter and binding array states are first-order reactions, the system dynamics can be described by a 2$N$ + 2 by 2$N$ + 2 transition matrix $A$ with elements $\left[a_{ij}\right]$.

The binding reaction (from $S_i$ to $S_{i+1}$) is modeled by:

$$a_{i,\,i+1} = a_{i+N+1,i+N+2} = k_i[Bcd] \text{ with } i \leq N. \tag{A1-2}$$

In *Equation A1-2*, $\left[Bcd\right] \sim e^{-x/\lambda}$ is the Bcd concentration, which can be calculated from the nuclei position $x$ and the decay length of the Bcd gradient $\lambda$.

The unbinding reaction (from $S_{i+1}$ to $S_i$) is modeled by:

$$a_{i+1,\,i} = a_{i+N+2,i+N+i} = k_{-i} \text{ with } 1 < i \leq N + 1. \tag{A1-3}$$

The activation of the promoter (from OFF state to ON state) is given by:

$$a_{i,i+N+1} = k_{ON}(i - 1, K) \text{ with } i \leq N + 1, \tag{A1-4}$$

In *Equation A1-4*, $K$ is the minimal number of bound Bcd molecules to the binding sites for transcription activation $k_{ON} \left(i < K, K\right) = 0$. With sufficient Bcd binding ($i \geq K$), the turning ON rate $k_{ON} \left(i, K\right)$ is modulated by $i$ the number of bound Bcd molecules.

The deactivation of the promoter (from ON state to OFF state) is given by:

$$a_{i+N+1,i} = k_{OFF} \text{ with } i \leq N + 1. \tag{A1-5}$$

To balance the transition matrix, we set the diagonal elements as:

$$a_{i,i} = \sum_j a_{i,j} \text{ with } j \neq i. \tag{A1-6}$$

With the transition matrix defined, we can calculate the probability vector at a given time $t$:

$$\bar{u} \left(t\right) = e^{-At}\bar{u}_0. \tag{A1-7}$$

The probability of the promoter to be ON is given by:

$$P_{ON} \left(t\right) = \bar{v}^\mathbf{T} \ \bar{u} \left(t\right), \tag{A1-8}$$

in which $\bar{v} = \left[v_i\right]$ is the emission vector specifying the ON state with $v_i = 1$ with $i > N + 1$ and 0 otherwise.

## Appendix 2

### Fitting the models of transcription regulation by Bcd

We fit the models of Bcd binding/unbinding to binding sites and activation of transcription, each with a different value of the Bcd gradient decay length $\lambda$, to the transcription dynamics by the synthetic reporters (*Figure 2* and *Figure 2—figure supplement 2*) and hb-P2 reporter (*Figure 5*).

### Data

The data used for the fitting is the fraction of active loci $P_{Spot}(j, t, x)$ at a given time $t$ and position $x$ along the AP axis (observed from –30 %EL to 20 %EL) in the $j^{th}$ embryo (of total $n$ embryos). $t$ ranges from 600 to 800 s into nc13 for synthetic reporters and from 0 s to 800 s into nc13 for hb-P2.

We discretize the time axis to increments of 10 s and nuclei position to increments of 1 %EL.

### Parameter constraints

#### Number of binding sites $N$

The model used in the fitting is described in the Results section and *Figure 2 (D–E)* of the main text. The number of Bcd binding sites is $N = 6$ for B6, H6B6, Z2B6 and hb-P2 (*Driever and Nüsslein-Volhard, 1988*), $N = 9$ for B9 and $N = 12$ for B12.

#### Bcd search rate constant for a target binding site $k_b$

The Bcd gradient follows an exponential gradient with a predefined decay length $\lambda$:

$$\left[Bcd\right] = c_A e^{-(x-x_A)/\lambda}, \tag{A2-1}$$

with $c_A$ chosen to be 140 nM or 84 molecules/µm³ (*Abu-Arish et al., 2010*). $x_A$ , which is 15 %EL, is the nuclei position where the Bcd gradient plateaus (*Gregor et al., 2007b*; *Houchmandzadeh et al., 2002*; *Liu et al., 2013*).

Unless otherwise specified, we assume that Bcd molecules independently diffuse in the 3D nuclear space in search for the target sites of the reporters and that the binding to the sites is diffusion limited. Therefore, the binding rate $k_i$ of Bcd to the binding site array depends on the number of free binding sites:

$$k_i = (N - i) k_b. \tag{A2-2}$$

Here, $k_b$ is the binding rate constant of individual Bcd molecule to a single target site and equal $\left(\tau_{bind}.\left[Bcd\right]\right)^{-1}$. The diffusion coefficient is taken to be D = 4.6 µm²/s and the target size a = 3 nm (*Abu-Arish et al., 2010*). Therefore, $k_b$ = 5.52 µm³/s.

#### Bcd unbinding rate constants $k_{-i}$

There are no constraints on the unbinding rate constants $k_{-i}$ of Bcd from the target binding array. However, for simplicity, we assume only two forms of Bcd binding cooperativity: bi-cooperativity and 6-cooperativity. It is found from *Tran et al., 2018b* that these two forms of cooperativity are sufficient to generate regulation functions of any order (Hill coefficient $H$ between 1 and 6) within the physical limit without extra energy expenditure (*Estrada et al., 2016*). Therefore, we use three free parameters $k_{-1}$ , $k_{-2}$ and $k_{-6}$ to describe $k_{-i}$. Assuming at state $S_{-(1<i<6)}$ , bound Bcd can unbind independently from the BS array, the intermediate binding rates are given by:

$$k_{-(1<i<6)} = ik_{-2}/2. \tag{A2-3}$$

#### Promoter switching rates $k_{ON}(S_i)$ and $k_{OFF}$

For the promoter activation and deactivation rates when bound by enough Bcd ($k_{ON}(S_i)$ and $k_{OFF}$), we adopt the scheme of the formation of transient "K-mer" (*Figure 2—figure supplement 3*) as it can explain the big fold change in the activation rate between $k_{ON}(S_6)$ , $k_{ON}(S_9)$ and $k_{ON}(S_{12})$. From *Supplementary file 6*, $K$ is chosen to be 3.

$$\mathrm{k}_{ON}\left(S_i\right) = \binom{i}{K} k_{ON}\left(S_K\right) = k_{ON}\left(S_6\right) \binom{i}{3} \Big/ \binom{6}{3} \qquad \text{(A2-4)}$$

In summary, for the fitting of synthetic reporters' patterns at steady state (*Figure 2*, panels E to G and *Figure 4*), the possible free parameters, with the corresponding ranges of values, are:

- Binding rate constants: $k_b$ with range [$e^{-20}$ /s, $e^{20}$ /s].
- The minimal number of Bcd to activate transcription $K$ with integer range (*Abu-Arish et al., 2010*; *Crocker and Ilsley, 2017*)
- Unbinding rate constants: $k_{-1}$, $k_{-2}$, $k_{-6}$ with range [$e^{-20}$ /s, $e^{20}$ /s]
- Activation and deactivation rate constants with $K$ bound Bcd molecules: $k_{ON}\left(S_K\right)$ and $k_{OFF}$ with range [0.001 / s, 100 / s]

For the fitting of hb-P2 pattern dynamics during nc13 (*Figure 5*), the free parameters, with the corresponding ranges of values, are:

- Unbinding rate constants: $k_{-1}$, $k_{-2}$, $k_{-6}$ with range [$e^{-20}$ s, $e^{20}$ s]
- Activation and deactivation rate constants with $K$ bound Bcd molecules: $k_{ON}\left(S_K\right)$ and $k_{OFF}$ with range [0.001 / s, 100 / s]

Additionally, when fitting hb-P2 pattern, we also fit the offset time $t_{offset}$ >0 as the first time Bcd molecules are allowed to interact with target binding sites following mitosis.

## Objective function

For each set of parameter $\phi = \left[k_b, k_{-1}, k_{-2}, k_{-6}, K, k_{ON}\left(S_6\right), k_{OFF}, t_{offset}\right]$, we calculate all the model rate constants $k_i$, $k_{-i}$, $k_{ON}\left(S_i\right)$, $k_{OFF}$ according to *Figure 2—figure supplement 3* and *Equation A2-2*, *Equation A2-3* and *Equation A2-4*. At a given nuclei position $x$, the Bcd concentration $\left[Bcd\right]$ is given by *Equation A2-1*. Given that all the reactions in the model (*Figure 2*) are first order, the system dynamics at nuclei position $x$ can be described by a transition matrix $A\left(x\right)$. We can predict the fraction of nuclei in the active state $P_{ON}\left(t, x\right)$ at any given time $t$ and nuclei position $x$ (see Appendix 1).

The objective function is given by the sum of squared errors from all $n$ embryos:

$$obj(\phi) = \sum_{j=1..n}\left(P_{ON}\left(t - t_{offset}, x\right) - P_{Spot}\left(j, t, x\right)\right)^2 \qquad \text{(A2-5)}$$

## Parameter estimation

The best fit parameter set $\bar{\phi}$ is given by minimizing the objective function:

$$\bar{\phi} = argmin_\phi\, obj\left(\phi\right) \qquad \text{(A2-6)}$$

In practice, we first generate ~2000 randomized initial values for the parameter set $\phi$ within the preset value ranges. For each initial value, we perform zeroth-order minimization using simplex search method with MATLAB's *fminsearch* function (*Lagarias et al., 1998*) to find the local minima. The best fit parameter set $\bar{\phi}$ is set corresponding to smallest local minimum.

# Appendix 3

## Evidence for synergistic activation between bound Bcd molecules

In the anterior region, we assume the binding array is always fully bound by Bcd. Thus, in our model, for a synthetic reporter $B_N$ with N Bcd sites ($N = 3,6,9,12$), the binding array state spends most of the time at state $S_N$. Because the MS2 stem-loops are placed at the 3' end of the transcribed sequence, we assume a negligible travel time of transcribing RNAP from the beginning of the first stem-loop to the terminator site (*Fukaya et al., 2017*). Thus, we approximate the probability of the promoter being in the ON state with the fraction of MS2 loci active time (bright MCP-GFP spots observed) at steady state:

$$P_{Spot}\left(B_N\right) = P_{ON}\left(S_N\right) = \frac{k_{ON}\left(S_N\right)}{k_{ON}\left(S_N\right)+k_{OFF}} \quad \text{(A3-1)}$$

We can calculate the activation rate $k_{ON}\left(S_N\right)$ as:

$$k_{ON}\left(S_N\right) = \left(k_{ON}\left(S_N\right) + k_{OFF}\right) P_{Spot}\left(B_N\right) \quad \text{(A3-2)}$$

$$k_{ON}\left(S_N\right) = k_{OFF} \frac{P_{Spot}\left(B_N\right)}{1-P_{Spot}\left(B_N\right)} \quad \text{(A3-3)}$$

With $P_{Spot}\left(B_N\right)$ observed from the data (N = 6, 9, 12), the fold change in the activation rates between B6, B9 and B12 is found to be:

$$\frac{k_{ON}\left(B9\right)}{k_{ON}\left(B6\right)} = \frac{P_{Spot}\left(B9\right)}{P_{Spot}\left(B6\right)}\frac{1-P_{Spot}\left(B6\right)}{1-P_{Spot}\left(B9\right)} = \frac{0.8*\left(1-0.47\right)}{\left(1-0.8\right)*0.47} \approx 4.5 \quad \text{(A3-4)}$$

$$\frac{k_{ON}\left(B12\right)}{k_{ON}\left(B6\right)} = \frac{P_{Spot}\left(B12\right)}{P_{Spot}\left(B6\right)}\frac{1-P_{Spot}\left(B6\right)}{1-P_{Spot}\left(B12\right)} = \frac{0.8*\left(1-0.47\right)}{\left(1-0.8\right)*0.47} \approx 4.5 \quad \text{(A3-5)}$$

This fold change between $k_{ON}\left(S_9\right)$ to $k_{ON}\left(S_6\right)$, which is three times greater than the ratio of the Bcd BS numbers between B9 and B6, argues against independent activation of transcription by individual bound Bcd TF, where the fold change scales with the number of BS ($K = 1$ and $k_{ON}\left(S_N\right) = N\,k_{ON}\left(S_1\right)$). Therefore, bound Bcd molecules are likely to cooperate with each other to activate transcription.

We considered various schemes of cooperative activation by bound Bcd and calculated the fold change in $k_{ON}$ for different values of $N$ and $K$ (*Figure 2—figure supplement 3* and *Supplementary file 6*). Among the schemes considered, the fold change in $k_{ON}$ from $S_6$ and $S_9$ was achieved only when bound Bcd can randomly form transient "$K$-mers" capable of activating transcription, and when $K$ is between 3 and 6.

However, the schemes with transient "$K$-mers" cannot explain the similar $P_{Spot}$ value observed in the anterior region for B9 and B12 reporters. One possibility is that Bcd molecules bound to the distal BS in B12 are too far from the TSS to activate transcription and thus $k_{ON}\left(S_{12}\right) = k_{ON}\left(S_9\right)$. Another explanation is that $P_{Spot} \approx 0.8$ is the upper limit imposed by inherent bursty dynamics of transcription even when the activation by bound Bcd is instantaneous ($k_{ON}\left(S_{12}\right), k_{ON}\left(S_9\right) \gg k_{ON}\left(S_6\right)$). In any case, the value of $\frac{k_{ON}\left(S_9\right)}{k_{ON}\left(S_6\right)} \sim 4.5$ calculated from the data should represent a lower bound for the value of the fold change in the activation rates. Thus, our conclusions regarding the synergistic activation by bound Bcd molecules still hold.

Note that, in the alternative model, where the Bcd-DNA complex $S_i$ regulates the deactivation rate $k_{OFF}\left(S_i\right)$ but not the activation rate $k_{ON}$. The ratio between $k_{ON}$ and $k_{OFF}\left(S_i\right)$ still holds, as in *Equation A3-4* and *Equation A3-5*. The fold change in the deactivation rates between B6, B9 and B12 is:

$$\frac{k_{OFF}\left(B9\right)}{k_{OFF}\left(B6\right)} = \frac{P_{Spot}\left(B6\right)}{P_{Spot}\left(B9\right)}\frac{1-P_{Spot}\left(B9\right)}{1-P_{Spot}\left(B6\right)} = \frac{0.47*\left(1-0.8\right)}{\left(1-0.47\right)*0.8} \approx 0.22 \quad \text{(A3-6)}$$

$$\frac{k_{OFF}\left(B12\right)}{k_{OFF}\left(B6\right)} = \frac{P_{Spot}\left(B6\right)}{P_{Spot}\left(B12\right)}\frac{1-P_{Spot}\left(B12\right)}{1-P_{Spot}\left(B6\right)} = \frac{0.47*\left(1-0.8\right)}{\left(1-0.47\right)*0.8} \approx 0.22 \quad \text{(A3-7)}$$

## Appendix 4

## Calculating the shift in pattern along AP axis

In this section, we present a framework to quantify the shift of the MS2 expression patterns along the AP axis from Bcd-2x to Bcd-1x flies.

We define two random variables $F$ and $G$ as the gene expression level in Bcd-2X and Bcd-1X embryos, respectively. Given that the experiments in Bcd-2X and Bcd-1X are independent, the probability distribution of $F$ and $G$ are independent. $X$ and $S$ are the nuclei position in Bcd-2X and Bcd-1X respectively, and are described by a uniform distribution from –50 %EL to 50 %EL. At any given position $x$ and $s$ in the embryo, $P_{F|X}\left(F|X=x\right)$ and $P_{G|S}\left(G|S=s\right)$ are assumed to be Gaussian distributed, with the mean and standard deviation equal to the mean and standard error of expression levels (fraction of expressing nuclei (*Figure 1D*).

We call $R$ the nuclei position in Bcd-1X that has the same expression level as in the nuclei at position $X$ in Bcd-2X. The conditional distribution $P_{R|X}$ (R|X = x) is given by:

$$
\begin{aligned}
P_{R|X}\left(R=s|X=x\right) &= \int_{-\infty}^{\infty}\int_{-\infty}^{\infty} df\,dg\, P_{FGS|X}\left(f,g,s|x\right)\,\delta\left(f-g\right)\\
&= \int_{-\infty}^{\infty}\int_{-\infty}^{\infty} df\,dg\, P_{F|X}\left(f|x\right) P_{G|X}\left(g|s\right) P_{S}\left(s\right)\,\delta\left(f-g\right).\\
&= \int_{\infty}^{\infty} df\, P_{F|X}\left(f|x\right) P_{G|S}\left(f|s\right) P_{S}\left(s\right)
\end{aligned}
\tag{A4-1}
$$

In *Equation A4-1*, $\delta\left(f-g\right)$ is a Dirac delta function taking non-zero value only when the expression level in Bcd-2X and Bcd-1X are equal $f=g$. As the nuclei position $S$ is uniformly distributed from –50 %EL to 50 %EL, $P_{S}\left(s\right)$ is constant and equal to $\left(\%EL\right)^{-1}$.

We denote by $\mu_{f}\left(x\right)$ and $\mu_{g}\left(s\right)$ the mean, $\sigma_{f}\left(x\right)$ and $\sigma_{g}\left(s\right)$ the standard deviation of $P_{F|X}\left(F|X=x\right)$ and $P_{G|S}\left(G|S=s\right)$ respectively.

$$
\begin{aligned}
P_{F|X}\left(f|x\right) &= \frac{1}{\sqrt{2\pi\sigma_{f}^{2}}}e^{-\frac{\left(f-\mu_{f}(x)\right)^{2}}{2\sigma_{f}^{2}(x)}}\\
P_{G|S}\left(f|s\right) &= \frac{1}{\sqrt{2\pi\sigma_{g}^{2}}}e^{-\frac{\left(f-\mu_{g}(s)\right)^{2}}{2\sigma_{g}^{2}(s)}}
\end{aligned}
\tag{A4-2}
$$

*Equation A4-2* can be rewritten as (suppressing the explicit x and s dependence):

$$
P_{R|X}\left(R=s|X=x\right)\ \sim\ \int_{\infty}^{\infty} df\frac{1}{\sqrt{2\pi 2\pi\sigma_{f}^{2}\sigma_{g}^{2}}}e^{-\frac{\left(f-\mu_{f}\right)^{2}}{2\sigma_{f}^{2}}-\frac{\left(f-\mu_{g}\right)^{2}}{2\sigma_{g}^{2}}}=e^{-C/2}
\tag{A4-3}
$$

with the term $C$ given by:

$$
C=\frac{1}{\sigma_{f}^{2}+\sigma_{g}^{2}}\left(\mu_{f}-\mu_{g}\right)^{2}+\ln\left(2\pi\left(\sigma_{f}^{2}+\sigma_{g}^{2}\right)\right)
\tag{A4-4}
$$

From *Equation A4-3* and *Equation A4-4*, we can analytically calculate the probability $P_{R|X}\left(s|x\right)$ for each pair of position $x$ and $s$ from the mean and standard error of gene expression level at a given position.

The probability distribution of the shift $\Delta\left(x\right)$ from position $x$ to position $P_{R|X}\left(R|X=x\right)$ is given by:

$$
P_{\Delta\left(x\right)}\left(\Delta|x\right)=P_{R|X}\left(R=x+\Delta|x\right)
\tag{A4-5}
$$

Given this probability distribution $P_{\Delta\left(x\right)}\left(\Delta\right)$ as a function of $x$, we can find a constant value of the shift $\Delta\left(x\right)=\widetilde{\Delta}$ that best describes the observed shift for all positions $x$ within $\epsilon X$:

$$
\widetilde{\Delta}=\arg\max_{\Delta}\int_{\epsilon X}\mathrm{p}\left(\Delta\left(x\right)=\Delta\right)dx
\tag{A4-6}
$$

## Appendix 5

### Decay length of the Bicoid protein gradient

As indicated in *Supplementary file 3*, several studies aiming to quantitatively measure the decay length of the Bicoid protein gradient ended-up with different values ranging from 16.4 %EL up to 25 %EL. These differences can potentially be attributed to different methods of detection (antibody staining on fixed samples *vs* fluorescent measurements on live sample) or to the type of protein detected (endogenous Bicoid *vs* fluorescently tagged). If measurements using antibody staining on fixed sample might suffer from variability in fixation efficiency among different embryos, each embryo is likely to be homogeneously fixed. Thus, this source of experimental variability should rather impact measurements of the "absolute" value of the Bcd concentration at the anterior ($c_A$ in *Equation A2-1* above) with a moderate impact on the decay length if background issues have been properly handled. In contrast, if live measurements from fluorescently tagged gradients solved the issue of variability among embryos and provide a more accurate measurement of $c_A$ , they suffer from other potential biases which might directly impact the decay length of the gradient. These biases include the delay in maturation time of the fluorescent tag (the newly synthesized proteins at the anterior will be less fluorescent than the more mature proteins which had time to diffuse away from the pole) or the presence of the fluorescent tag in the fusion protein, which might affect its diffusion coefficient or half-life as compared to the wild-type protein. The issue of the fluorescent tag maturation time has been carefully analyzed in recent studies (*Durrieu et al., 2018*; *Liu et al., 2013*). Comparison of the decay length of a GFP-tagged Bcd gradient using either live fluorescent measurements or antibody staining on fixed embryos allowed Liu et al. to evaluate the amplitude of the bias due to the maturation time of the GFP fluorescent tag: the decay length of the Bcd-GFP gradient measured through the GFP fluorescence is 19.3% EL while it is only 16.4% EL when measured through a fluorescent immunostaining for Bcd (Fig. S4 in *Liu et al., 2013*). Surprisingly, the value of the Bcd-GFP decay length detected by immuno-staining with a Bcd antibody is 3.6% EL lower than the decay length evaluated with the same approach on the endogenous Bcd protein gradient (*Houchmandzadeh et al., 2002*). The reason for this difference is not discussed by the authors and remains unclear. It could be due to subtle differences in the experimental procedures used in the two studies. Another explanation is the nature of the protein detected in each study, which is the wild-type Bcd in the first study (*Houchmandzadeh et al., 2002*) and the Bcd-GFP fusion in the second study (*Liu et al., 2013*), as the presence of the fluorescent tag might modify the physical parameters of the protein (diffusion, half-life) and impact the decay length of the gradient as previously proposed (*Xu et al., 2015*).

Thus, the exact value of the Bcd protein gradient decay length is not known and that we only have measurements that put it in between 16% and 25% EL.

## Appendix 6

### Bicoid search time for its target sites and the Berg & Purcell limit

We recalculate the Bcd search time for its target site at the critical *hb* expression boundary ( ~ 45 %EL) with different values of the Bcd gradient decay length $\lambda$. The Bcd gradient follows an exponential gradient with a fixed concentration at the anterior $c_A$ (*Equation A2-1*). We assume that each Bcd molecule searches for its target sites via 3D diffusion in the nuclear space, with a diffusion coefficient $D$. Each target site has a size of $a$. If the binding of Bcd to the target site is diffusion limited, the Bcd search time for individual target sites $\tau_{bind}$ at a given position $x$ is given by:

$$t_{bind} \sim \left( Dac_A e^{-(x-x_A)/\lambda} \right)^{-1} \tag{A6-1}$$

From *Equation A6-1*, it can be seen that the search time $t_{bind}$ is longer with decreasing $\lambda$.

In practice, there are many uncertainties regarding the exact values of $c_A$, $D$ and $a$, making it difficult to estimate $t_{bind}$ (See *Supplementary file 4* for details).

If the target gene can sense the Bcd concentration via $N$=6 identical and independent binding sites, we calculate the minimum observation time $T$ it takes to achieve the prediction error $\delta[Bcd]/[Bcd]$ = 10% of the Bcd concentration at the hb-P2 boundary ($x$ = 45% EL) in the *Berg and Purcell, 1977*:

$$\frac{\delta[Bcd]}{[Bcd]} = \sqrt{\frac{1}{NTDac_A e^{-(x-x_A)/\lambda}}} \tag{A6-2}$$

$$T = \left( \frac{\delta[Bcd]}{[Bcd]} \right)^{-2} \frac{1}{NDac_A e^{-(x-x_A)/\lambda}} = \frac{16.7}{Dac_A e^{-30/\lambda}} \tag{A6-3}$$

We compare the target site search time $t_{bind}$ and the minimum required readout time $T$ with $\lambda$ equal 15 %EL and 20 %EL in *Supplementary file 5*. We consider different combinatory values of $c_A$, $D$ and $a$ found in *Supplementary file 4*.

