## [Editor Report]

In this paper, the authors use synthetic transcriptional enhancers to probe the roles of three transcription factors, Bicoid, Hunchback and Zelda, in specifying the production of a sharp, accurately placed gene expression boundary in early fruit fly embryos. They find that Bicoid, which is expressed in an anterior-posterior gradient, is sufficient on its own to generate a boundary in the same location as a wild-type fly, but combinatorial regulation by Hunchback and Zelda is needed to ensure the boundary forms quickly enough. They further combine their experimental observations with modeling to conclude that Bicoid exists in active and inactive forms, and that an equilibrium model captures the relevant behaviors, implying energy expenditure during the binding of transcription factors to DNA or RNA polymerase is theoretically unnecessary.

---

## [Decision Letter]

**Decision letter after peer review:**

Thank you for submitting your article "Synthetic deconstruction of *hunchback* regulation by Bicoid" for consideration by *eLife*. Your article has been reviewed by 3 peer reviewers, and the evaluation has been overseen by a Reviewing Editor and Naama Barkai as the Senior Editor. The following individuals involved in review of your submission have agreed to reveal their identity: Timothy Saunders (Reviewer #1); Justin Crocker (Reviewer #3).

The reviewers have discussed their reviews with one another, and the Reviewing Editor has drafted this to help you prepare a revised submission. While all the reviewers found the study interesting, they also unanimously agree that more data and clarifications are required to substantiate the claims of the manuscript. Please do address all the major comments in the reviews below in your revision.

*Reviewer #1 (Recommendations for the authors):*

– My major concern regards the conclusions for the final section on the activity gradient. In the Introduction it is stated: "[the Bcd gradient has] an exponential AP gradient with a decay length of ~ 20% egg‐length (EL)". While this was the initial estimate (Houchmandzadeh et al., Nature 2002), later measurements by the Gregor lab (see Supplementary Material of Liu et al., PNAS 2013) found that "The mean length constant was reduced to 16.5 {plus minus} 0.7%EL after corrections for EGFP maturation". The original measurements by Houchmandzadeh et al. had issues with background control, that also led to the longer measured decay length. In later work, Durrieu et al., MSB 2018, found a similar scale for the decay length to Liu et al. Looking at Figure 5, a value of 16.5%EL for the decay length is fully consistent with the activity and protein gradients for Bcd being similar. So, either (1) the authors need to provide convincing evidence that, for some reason, in their specific lines the Bcd gradient is longer than expected; or (2) they need to alter the conclusions of the final sections. Indeed, the evidence to me is interesting as it shows that the activity gradient and protein gradient actually are similar (e.g. 15% EL activity vs 16.5% EL protein) which is not, as the authors highlight, a priori obvious. Now, this small shift may well be significant statistically and it is a worthy point in its own right. However, the strength of the conclusions clearly do not match the known gradient and should be substantially toned down.

Related to the above, the abstract, intro and discussion all need parts to be rewritten to reflect more accurately the data interpretation. Given the current evidence presented in the manuscript, all conclusions on the activity gradient relative to the protein gradient are incorrect.

– While the videos are nice, it would be very helpful to show some of the actual data in the figures. Clearly annotated time points would significantly enhance the readability of the first section. The heatmaps are insufficient. This is also true for Figure 2A – show some real data to highlight the burstiness point.

– The axis scales are not helpful. I'm assuming that "0% EL" is meant to be the hb boundary. This is not defined in legend of figure 1, indeed there i simply says distance along embryo axis. Put 0% at one of the poles. This is convention and I see no reason to define 0% at a (somewhat arbitrary) position in the centre of the embryo. Please keep to standard nomenclature unless a very compelling reason. I think this also leads to confusion. E.g. line 242 states that the expression boundary is at 17.5% EL (Figure 3C), yet the results in this figure only extend to 10%EL. Please clarify.

– All of the experiments are performed in a background with the hb gene present. Does this impact on the readout, as the synthetic lines are essentially competing with the wild-type genes. What controls were done to account for this?

– Further, the activity of the synthetic reporters depend on the location of insertion. Erceg et al. PLoS Genetics 2014 showed that the same synthetic enhancer can have different readout depending on its genomic location. I'm aware that the authors use a landing site that appears to replicate similar hb kinetics, but did they try random insertion or other landing site? In short, how robust are their results to the specific local genome site? This should have been tested, especially given the boldly written conclusions from the work.

– Related to the above, it's also not obvious that readout is linear – i.e. as more binding sites are added, there could be cooperativity between binding domains. This may have been accounted for in the model but it is not clear to me how.

– It would be good in the Introduction/Discussion to give a broader perspective on the advantages and disadvantages of the synthetic approach to study gene regulation. The intro only discusses Tran et al., Yet, there is a strong history of using this approach, which has also helped to reveal some of the approaches shortcoming. E.g. Gertz et al., Nature 2009 and Sharon et al., Nature Biotechnology 2012. Again, I may have missed, but from my reading I cannot see any critical analysis of the pros/cons of the synthetic approach in development. This is necessary for a broad journal such as *eLife* so that readers are given clearer context.

*Reviewer #3 (Recommendations for the authors):*I think the authors should acknowledge more explicitly that Hb in the endogenous system acts in an autoregulatory fashion, which it does not do in the synthetic construct. This is relevant to two of the conclusions in particular:

1) I agree that Hb's effects on bursting would increase patterning speed. However, in the B6+Hb BS construct an additional contribution to patterning dynamics could come from the fact that the endogenous Hb boundary is itself dynamic and forms faster than B6 alone. Unless we know how the reporter construct would act with autoregulation orthogonal to Hb, we can't say for sure whether adding Hb BS to B6 gives B6 access to higher concentrations of activator in the anterior than would be expected at the same time point in the endogenous autoregulatory loop.

2) In lines 329-330 the authors state that "the effects of Zld and Hb are modeled implicitly by the kinetic rate constants". Is this valid? The Hb boundary is dynamic during the relevant time window, and because Hb in hb-p2 is autoregulatory even time-varying kinetic rates fixed to a measured dynamic trajectory for Hb concentration wouldn't be valid for modeling/drawing conclusions about the endogenous network.

I find the logscale visualization in Figure 4 unnecessarily involved when the reader simply needs to know the approximate size of the boundary shift. I would move the plots to the supplement.

---

## [Author Response]

Reviewer #1 (Recommendations for the authors):– My major concern regards the conclusions for the final section on the activity gradient. In the Introduction it is stated: "[the Bcd gradient has] an exponential AP gradient with a decay length of ~ 20% egg‐length (EL)". While this was the initial estimate (Houchmandzadeh et al., Nature 2002), later measurements by the Gregor lab (see Supplementary Material of Liu et al., PNAS 2013) found that "The mean length constant was reduced to 16.5 {plus minus} 0.7%EL after corrections for EGFP maturation". The original measurements by Houchmandzadeh et al. had issues with background control, that also led to the longer measured decay length. In later work, Durrieu et al., MSB 2018, found a similar scale for the decay length to Liu et al. Looking at Figure 5, a value of 16.5%EL for the decay length is fully consistent with the activity and protein gradients for Bcd being similar. So, either (1) the authors need to provide convincing evidence that, for some reason, in their specific lines the Bcd gradient is longer than expected; or (2) they need to alter the conclusions of the final sections. Indeed, the evidence to me is interesting as it shows that the activity gradient and protein gradient actually are similar (e.g. 15% EL activity vs 16.5% EL protein) which is not, as the authors highlight, a priori obvious. Now, this small shift may well be significant statistically and it is a worthy point in its own right. However, the strength of the conclusions clearly do not match the known gradient and should be substantially toned down.

The reviewer is right: several studies aiming to quantitatively measure the Bicoid protein gradient ended-up with quite different decay lengths.

A summary of the various decay lengths measured, and the method used for these measurements is given in Author response table 1.

**Author response table 1. sa2table1:** 

λ in µm	λ as % EL	Method of detection	Reference
100	~ 20	fixed embryos, antibody against the wt Bicoid protein	Houchmandzadeh et al., Nat, 2002
125	~ 25	live detection of the Bcd-eGFP fusion	Abu-Arish et al., Biophys, 2010
	19.3	live detection of the Bcd-eGFP fusion	Liu et al., PNAS 2013 (Figure S4)
	18.2	live detection of the Bcd-Venus fusion	Liu et al., PNAS 2013 (Figure S4)
	16.4	fixed embryos, Bcd antibody against Bcd-eGFP	Liu et al., PNAS 2013 (Figure S4)
100	20	Fixed embryos, antibodies against the wt Bicoid protein at nc13	Xu et al., Nature Methods 2015
89	~ 17.8	Live detection of tandem-fluorescent protein timer fused to Bcd	Durrieu et al., Mol Sys Biol 2018 (Page 7)

As indicated, these measurements are quite variable among the different studies and the differences can potentially be attributed to different methods of detection (antibody staining on fixed samples *vs* fluorescent measurements on live sample) or to the type of protein detected (endogenous Bicoid *vs* fluorescently tagged).

We agree with the reviewer that given these discrepancies, the exact value of the Bcd protein gradient decay length is not known and that we only have measurements that put it in between 16 and 25 % EL (see Author response table 1). Therefore, we agree that we should tone down the difference between the protein *vs* activity gradient and focus on the measurements of the effective activity gradient decay length allowed by our synthetic reporters. This allows us to revisit the measurement of the Hill coefficient of the transcription step-like response, which is based on the decay-length for the Bcd protein gradient, and assumed in previous published work to be of 20% EL (Gregor et al., Cell, 2007a; Estrada et al., 2016; Tran et al., PLoS CB, 2018). Importantly, the new Hill coefficient allows us to set the Bcd system within the limits of an equilibrium model.

As mentioned by the reviewer, it is possible that the decay length of the protein gradient measured using antibody staining (Houchmandzadeh et al., Nature, 2002) was not correct due to background controls. Such measurements were also performed in Xu et al., (2015) which agree with the original measurements (Houchmandzadeh et al., Nature 2002). As indicated in Author response table 1, all the other measurements of the Bcd protein gradient decay length were done using fluorescently tagged Bcd proteins and we cannot exclude the possibility the wt *vs* tagged protein might have different decay lengths due to potentially different diffusion coefficients or half-lives. Before drawing any conclusion on the exact value of the endogenous Bcd protein gradient decay length, it is essential to measure it again in conditions that correct for the background issues for immuno-staining as it was done in Liu et al., PNAS, 2013 for the Bcd-eGFP protein. In this study, the authors only measured the decay length of the Bcd fusion protein using immuno-staining for the Bcd protein. Unfortunately, in this study, the authors did not measure again the decay length of the endogenous Bcd protein gradient using immuno-staining and the same procedure for background control. Therefore, they do not firmly exclude the possibility that the endogenous *vs* tagged Bcd proteins might have different decay length.

We thank the reviewer for his comment which helped us to clarify the message:

– In the result section, we only focus on the decay length measurements of the Bcd activity gradient using the Bcd-only reporters, putting the emphasis on the fact that the decay length measured is short and places back the Bcd system within the physical limits of an equilibrium model.

– In addition, as there is clearly an issue for the measurements of the Bcd protein gradient, we added an Appendix (Appendix 5, lines 1111 to 1142) and a Table (Supplementary File 3) describing the various decay length measured for the Bcd or the Bcd-fluorescently tagged protein gradients from previous studies.

– Finally , in the discussion (lines 482 to lines 490), together with the possibility that there might be a protein *vs* activity gradient (as we originally proposed and believe is still a valid possibility), we also discuss the alternative possibility proposed by the reviewer which is that the protein *vs* activity gradients have the same decay lengths but that the decay length of the Bcd protein gradient was potentially not correctly evaluated.

Related to the above, the abstract, intro and discussion all need parts to be rewritten to reflect more accurately the data interpretation. Given the current evidence presented in the manuscript, all conclusions on the activity gradient relative to the protein gradient are incorrect.

The sentences related to this aspect have been rephrased to be more accurate: in most cases (except in the discussion), we only focus on the decay length measurements of the Bcd activity gradient using the Bcd-only reporters, putting the emphasis on the fact that the decay length measured is short and places back the Bcd system within the physical limits of an equilibrium model.

The modifications in the main text concerning this point are :

– Introduction: line 99

– Result section: everything concerning the decay length of the Bcd protein gradient has been removed from line 283 to the end of the section. The title of the section line 283 has been changed in “A Bicoid-Activity gradient with a short decay length”

– Discussion: lines 482 to 490

– While the videos are nice, it would be very helpful to show some of the actual data in the figures. Clearly annotated time points would significantly enhance the readability of the first section. The heatmaps are insufficient. This is also true for Figure 2A – show some real data to highlight the burstiness point.

We completely agree with this comment which will help the reader to grasp even more what the MS2 videos bring.

We added a new figure (Figure 2—figure supplement 1) to highlight these aspects. In this figure, we show :

– A fluorescent trace (intensity of a locus as function of time) along a whole nc13 (Panel A) and series of snapshots at chosen times for this specific trace (Panel B) of the B6 reporter which is the most fluctuating one.

– Kymographs of traces to highlight burstiness of all the reporters used: these are traces all obtained in nuclei located at the anterior (where Bcd is supposed to be in excess). 20 traces for each reporter used in this study are shown Panel C.

– The axis scales are not helpful. I'm assuming that "0% EL" is meant to be the hb boundary. This is not defined in legend of figure 1, indeed there i simply says distance along embryo axis. Put 0% at one of the poles. This is convention and I see no reason to define 0% at a (somewhat arbitrary) position in the centre of the embryo. Please keep to standard nomenclature unless a very compelling reason. I think this also leads to confusion. E.g. line 242 states that the expression boundary is at 17.5% EL (Figure 3C), yet the results in this figure only extend to 10%EL. Please clarify.

As requested by the reviewer, the anterior pole has been set at 0% EL and the posterior at 100% EL in all figures, figure supplements and supplementary File 2. In line 270, we have clarified that the shift of the expression boundary (not the absolute position itself) between Z2B6 and B6’s boundary position is *17.5 %EL*.

– All of the experiments are performed in a background with the hb gene present. Does this impact on the readout, as the synthetic lines are essentially competing with the wild-type genes. What controls were done to account for this?

We agree with the reviewer that this concern might be particularly relevant at the *hb* boundary where a nucleus has been shown to only contain ~ 700 Bicoid molecules (Gregor et al., Cell, 2007b). However, ~1000 Bicoid binding regions have been identified by ChIP seq experiments in nc14 embryos (Hannon et al., *eLife*, 2017) and given that several Bcd binding sites are generally clustered together in a Bcd region, the number of Bcd binding sites in the fly genome is likely larger than 1000. It is much greater than the number of Bicoid binding sites in our synthetic reporters. Therefore, we think that it is unlikely that adding the synthetic reporters (which in the case of B12 only represents at most 1/100 of the Bcd binding sites in the genome) will severely alter the competition for Bcd binding between the other Bcd binding sites in the genome. Additionally, the insertion of a BAC spanning the endogenous *hb* locus with all its Bcd-dependent enhancers did not affect (as far as we can tell) the regulation of the wild-type gene (Lucas, Tran et al., 2018).

We have added a sentence concerning this point in the main text (lines 110 to 113).

– Further, the activity of the synthetic reporters depend on the location of insertion. Erceg et al., PLoS Genetics 2014 showed that the same synthetic enhancer can have different readout depending on its genomic location. I’m aware that the authors use a landing site that appears to replicate similar hb kinetics, but did they try random insertion or other landing site? In short, how robust are their results to the specific local genome site? This should have been tested, especially given the bo’dly written conclusions from the work.

This concern of the reviewer has been tested and is addressed Figure 1—figure supplement 1 where we compare two random insertions of the hb-P2 transgene (on chromosome II and III; Lucas, Tran et al., 2018) and the insertion at the VK33 landing site that was used for the whole study. As shown Figure 1—figure supplement 1, the dynamics of transcription (kymographs) are very similar. In the main text, the reference Figure 1—figure supplement 1 is found in the result section (line 107) and in the Materials and methods section (line 529).

– Related to the above, it’s also not obvious that readout is linear – i.e. as more binding sites are added, there could be cooperativity between binding domains. This may have been accounted for in the model but it is not clear to me how.

The reviewer is totally correct. It is clear from our data that readout is not linear: comparing (increase of 1.5 X in the number of BS) B6 with B9 leads to a 4.5 X greater activation rate and this argues against independent activation of transcription by individual bound Bcd TF. There is almost no impact of adding 3 more sites when comparing B9 to B12 (even though it corresponds to an increase of 1.33 X in the number of BS).

This issue has been rephrased in the main text (lines 205 to 208) and is further developed for the modeling aspects in the Appendix 3 and Figure 2—figure supplement 2. It is also discussed in the second paragraph of the discussion (lines 392 to 395).

– It would be good in the Introduction/Discussion to give a broader perspective on the advantages and disadvantages of the synthetic approach to study gene regulation. The intro only discusses Tran et al. Yet, there is a strong history of using this approach, which has also helped to reveal some of the approaches shortcoming. E.g. Gertz et al., Nature 2009 and Sharon et al., Nature Biotechnology 2012. Again, I may have missed, but from my reading I cannot see any critical analysis of the pros/cons of the synthetic approach in development. This is necessary for a broad journal such as eLife so that readers are given clearer context.

One sentence was added in the introduction concerning this point (lines 80 to 83).

A short review concerning the synthetic approach in development has also been added at the beginning of the discussion (lines 359 to 371).

Reviewer #3 (Recommendations for the authors):I think the authors should acknowledge more explicitly that Hb in the endogenous system acts in an autoregulatory fashion, which it does not do in the synthetic construct. This is relevant to two of the conclusions in particular:1) I agree that Hb's effects on bursting would increase patterning speed. However, in the B6+Hb BS construct an additional contribution to patterning dynamics could come from the fact that the endogenous Hb boundary is itself dynamic and forms faster than B6 alone. Unless we know how the reporter construct would act with autoregulation orthogonal to Hb, we can't say for sure whether adding Hb BS to B6 gives B6 access to higher concentrations of activator in the anterior than would be expected at the same time point in the endogenous autoregulatory loop.

We totally agree that the endogenous Hb boundary formed thanks to the endogenous autoregulatory loop contributes to the dynamics of the H6B6 construct. In our synthetic approach, none of our reporters express the Hb protein, and thus we focused our work on understanding the role of each BS on transcription in the context of the endogenous autoregulation. Thus, comparing H6B6 and B6 let us understand the role of Hb BS on transcription in the same context of Hb protein dynamics. We did not focus our work on the combined effect of transcription and autoregulation, which would require an orthogonal system as noted by reviewer #3.

2) In lines 329-330 the authors state that "the effects of Zld and Hb are modeled implicitly by the kinetic rate constants". Is this valid? The Hb boundary is dynamic during the relevant time window, and because Hb in hb-p2 is autoregulatory even time-varying kinetic rates fixed to a measured dynamic trajectory for Hb concentration wouldn't be valid for modeling/drawing conclusions about the endogenous network.

This assumption is clearly valid for Zelda which is homogeneously distributed all along the AP axis. For Hb, as noted by the reviewer, it is clear from our study that the *hb* transcription boundary is dynamic during the relevant time window. However, a time delay is expected concerning the corresponding increase in the nuclear concentration of the Hb protein (the activator) : the system requires nuclear export of the *hb* mRNA, its translation and the nuclear import of the translated Hb protein to affect the activity of the promoters. Given the extremely fast time scale of boundaries establishment (3 min), we don’t think that there would be a significant effect on patterning dynamics coming from endogenous *hb* transcribed at the same time over this time scale. This time delay has also to be considered in the context of the very dynamic nuclear division process. It is thus very likely that the nuclear Hb protein at work at nc13 was produced from mRNA expressed at earlier nuclear cycles. Therefore, we assumed that the Hb protein expression domain is well delimited by the position of the hb-P2 transcription boundary and that the amount of Hb protein expressed is constant both in its expression domain and along the period of the interphase nuclear cycle during which transcription occurs. As the boundaries of B6 and H6B6 are more anterior than the boundary of hb-P2, these boundaries are localized in a domain where the Hb protein is at its maximum and at a plateau.

I find the logscale visualization in Figure 4 unnecessarily involved when the reader simply needs to know the approximate size of the boundary shift. I would move the plots to the supplement.

We agree with the reviewer that as such this part of the Figure 4 was not very convincing nor did it add more than the measurements of the shifts and the fits with the position extracted from the model assuming the calculated decay length. We think that this was due to the fact that the logscale panels C and E of Figure 4 correspond to individual plots from the B9 (C) or the hb-P2 (E) reporters. Since the first submission of this manuscript, we were able to collect additional data for the B6 and Z2B6 reporters in the same experimental context (i.e. 1X vs 2X embryos using the *Δbcd* strain). It became thus possible to merge the logscale figures for each of the 4 reporters. As these reporters have different boundary position along the AP axis and thus different threshold values of Bcd concentration for positioning their boundary, the determination of the effective decay length is obtained on a wider segment of the AP axis. The merged logscale has a wider horizontal region confirming that the effective activity gradient of Bcd is exponential at least in this part of the AP axis.

We have added this new merged figure in Figure 4 (panel E) as well as the graphs obtained for the two additional reporters B6 (panel B) and Z2B6 (panel D). Of note, H6B6 cannot be used in this assay as the expression of the Hb protein varies in the 1X vs 2X *Δbcd* genetic context leading to a different situation as for the other reporters which are not themselves responding to Hb itself.